∂ | Open Peer Review | Environmental Microbiology | Research Article

# *Neobacillus driksii* sp. nov. isolated from a Mars 2020 spacecraft assembly facility and genomic potential for lasso peptide production in *Neobacillus*

Asif Hameed,[1] Francesca McDonagh,[2] Pratyay Sengupta,[3,4,5] Georgios Miliotis,[2,6] Shobhan Karthick Muthamilselvi Sivabalan,[3] Lukasz Szydlowski,[7] Anna Simpson,[8,9] Nitin Kumar Singh,[8] Punchappady Devasya Rekha,[1] Karthik Raman,[4,10] Kasthuri Venkateswaran[8]

**ABSTRACT** During microbial surveillance of the Mars 2020 spacecraft assembly facility, two novel bacterial strains, potentially capable of producing lasso peptides, were identified. Characterization using a polyphasic taxonomic approach, whole-genome sequencing and phylogenomic analyses revealed a close genetic relationship among two strains from Mars 2020 cleanroom floors (179-C4-2-HS, 179-J1A1-HS), one strain from the Agave plant (AT2.8), and another strain from wheat-associated soil (V4I25). All four strains exhibited high 16S rRNA gene sequence similarity (>99.2%) and low average nucleotide identity (ANI) with *Neobacillus niacini* NBRC 15566$^T$, delineating new phylogenetic branches within the genus. Detailed molecular analyses, including *gyr*B (90.2%), ANI (86.4%), average amino acid identity (87.8%) phylogenies, digital DNA–DNA hybridization (32.6%), and percentage of conserved proteins (77.7%) indicated significant divergence from *N. niacini* NBRC 15566$^T$. Consequently, these strains have been designated *Neobacillus driksii* sp. nov., with the type strain 179-C4-2-HS$^T$ (DSM 115941$^T$ = NRRL B-65665$^T$). *N. driksii* grew at 4°C to 45°C, pH range of 6.0 to 9.5, and 0.5% to 5% NaCl. The major cellular fatty acids are iso-C$_{15:0}$ and anteiso-C$_{15:0}$. The dominant polar lipids include diphosphatidylglycerol, phosphatidylglycerol, phosphatidylethanolamine, and an unidentified aminolipid. Metagenomic analysis within NASA cleanrooms revealed that *N. driksii* is scarce (17 out of 236 samples). Genes encoding the biosynthesis pathway for lasso peptides were identified in all *N. driksii* strains and are not commonly found in other *Neobacillus* species, except in 7 out of 26 recognized species. This study highlights the unique metabolic capabilities of *N. driksii*, underscoring their potential in antimicrobial research and biotechnology.

**IMPORTANCE** The microbial surveillance of the Mars 2020 assembly cleanroom led to the isolation of novel *N. driksii* with potential applications in cleanroom environments, such as hospitals, pharmaceuticals, semiconductors, and aeronautical industries. *N. driksii* genomes were found to possess genes responsible for producing lasso peptides, which are crucial for antimicrobial defense, communication, and enzyme inhibition. Isolation of *N. driksii* from cleanrooms, Agave plants, and dryland wheat soils, suggested niche-specific ecology and resilience under various environmentally challenging conditions. The discovery of potent antimicrobial agents from novel *N. driksii* underscores the importance of genome mining and the isolation of rare microorganisms. Bioactive gene clusters potentially producing nicotianamine-like siderophores were found in *N. driksii* genomes. These siderophores can be used for bioremediation to remove heavy metals from contaminated environments, promote plant growth by aiding iron uptake in agriculture, and treat iron overload conditions in medical applications.

**Peer Reviewer** Srikrishna Subramanian, Institute of Microbial Technology (CSIR), Chandigarh, India

Address correspondence to Karthik Raman, kraman@iitm.ac.in, or Kasthuri Venkateswaran, kjvenkat1955@gmail.com.

Asif Hameed, Francesca McDonagh, and Pratyay Sengupta contributed equally to this article. Line of authorship is based on the alphabetical order of their names.

The authors declare no conflict of interest.

See the funding table on p. 22.

KEYWORDS *Neobacillus driksii*, spore-former, whole-genome sequencing, lasso peptides, Mars 2020, cleanrooms

*N*eobacillus species are emerging as a critical group of microorganisms due to their versatile roles in environmental sustainability, ecological balance, and scientific exploration. The bacterial genus *Neobacillus*, classified within the family *Bacillaceae* and order *Bacillales*, encompasses a group of rod-shaped microorganisms that exhibit Gram-positive or Gram-variable staining characteristics. Phylogenetic analyses have revealed distinct genetic differences between *Neobacillus* and the more comprehensive *Bacillus* genus, thereby justifying its recognition as a separate and distinct genus (1). As of June 2024, there are 26 species within this genus that form a monophyletic branch based on specific conserved DNA/protein sequences.

In a microbial surveillance study associated with the NASA Mars 2020 mission's spacecraft assembly environments, 110 strains were isolated after cleanroom samples were subjected to a heat-shock (80°C; 15 min) procedure and were cultured aerobically as per the NASA standard spore assay (2). Subsequently, whole genome sequencing (WGS) was performed on these strains (3), and the average nucleotide identity (ANI) analysis indicated that two of the strains did not match (<87% ANI similarity) with any known bacterial species. Phylogenies based on genes encoding 16S rRNA and DNA gyrase subunit B (*gyrB*) placed them in the genus *Neobacillus*. During genome mining of all *Neobacillus* strains (*n* = 113 genomes) available in the public domain, two additional strains closely related to two Mars 2020 isolates based on WGS were identified. Although the physiological and biochemical characteristics of these strains were not evaluated, a comparative genome analysis was conducted on these four strains alongside other *Neobacillus* species.

The aim of this study was to identify the taxonomic affiliation through traditional and genomic analyses. In addition to the WGS phylogeny, conserved marker genes encoding 16S rRNA, DNA gyrase subunit B (*gyrB*), imidazole glycerol phosphate synthase subunit (*hisH*)*,* and ATP phosphoribosyltransferase regulatory subunit (*hisZ*) were analyzed, and the phylogenetic affiliations of the strains within spore-forming genera were delineated (1). The *hisH* and *hisZ* genes were chosen since these genes were present in all *Neobacillus* species. A second objective was to determine the abundance of the newly discovered *Neobacillus* species on the surfaces of the spacecraft assembly facility (SAF) cleanrooms where the Mars 2020 mission had been assembled. A total number of 236 pair-end shotgun samples (116 samples treated with propidium monoazide [PMA] and 120 samples without any PMA treatment), were examined to assess the occurrence and prevalence of the novel species. Detailed genetic profiles of the species were created, and distinct phenotypic traits were predicted.

A third objective was to generate functional characterization of the newly discovered *Neobacillus* species. One of the predictive analyses carried out during this study identified unique compounds, such as lasso peptide, potentially produced by the newly described *Neobacillus* species. Lasso peptides have garnered special interest recently due to their unique structural and functional properties. These peptides can form cyclic elements, such as loops and knots, which are associated with their antimicrobial, antiviral, antitumor, and enzyme-inhibiting properties (4). The rotaxane slipknot conformation renders lasso peptides resistant to thermal and enzymatic degradation. Until recently, lasso peptide-associated genes have been exclusive to Gram-negative bacteria; however, the presence of novel, lasso peptide-producing biosynthetic gene cluster (BGC) in Gram-positive *Paenibacillus dendritiformis* C454 has been reported, hence referred to as paeninodin (4). The paeninodin-producing BGC contains a novel type of kinase, specific for C- terminal serine residue. The functional analyses provided insights into how rare bacterial species, such as *Neobacillus* species, thrive and interact in the stringent, nutrient-scarce conditions of the SAF cleanroom environments, showcasing their resilience and ecological importance.

In oligotrophic cleanroom environments, rare bacterial species play a crucial role in biodegradation, breaking down complex organic pollutants through their enzymatic capabilities (5). Their ability to decompose substances resistant to decay facilitates the recycling of essential nutrients (6). In bioremediation, microbial species are particularly effective in cleaning up contaminated sites, including those polluted with heavy metals or organic waste, making them invaluable for restoring damaged ecosystems (7). The adaptability and ability of *Neobacillus* species to thrive in diverse environments—ranging from wheat grown soil (V4I25 strain) and plants (AT2.8 strain from Agave plants) to oligotrophic cleanrooms (Mars 2020 strains)—demonstrate their ecological versatility (8).

## RESULTS

### Ecology

In the JPL-SAF cleanroom, amidst the assembly of the Mars 2020 mission components on March 30, 2016, microbial analysis yielded 300 CFU/m$^2$ when grown in tryptic soy agar (TSA) incubated at 30°C and the same sample after heat shocking (80°C; 15 min) showed presence of 24 CFU/m$^2$, constituting about 17%, and formed spores. The WGS was carried out on 17 heat-shocked strains and 13 non-heat-shocked isolates to determine their phylogenetic lineages (3). Aside from the newly characterized *Neobacillus* species (strain 179-C4-2-HS$^T$), the analysis identified two strains of *Bacillus subtilis*, and one strain each of *Virgibacillus pantothenticus*, *Neobacillus niacini*, *Priestia megaterium*, and *Brevibacillus parabrevis* among the heat-shocked group. Ten additional heat-shocked isolates were grouped under the genus *Brevibacillus* without species identification. From the non-heat-shock group, five strains were found to be *Bacillus safensis*, one *Bacillus pumilus*, two *Micrococcus luteus*, and one *P. megaterium*. Four strains remained unidentified, falling within the genera *Brevibacillus, Paenibacillus, Priestia,* and *Microbacterium*. In a separate sampling on July 26, 2016, 100 CFU/m$^2$ cultivable bacteria were detected at the same cleanroom floors when grown in TSA medium, with 29 CFU/m$^2$ (~36%) surviving heat shock. When WGS was performed on four heat-shocked strains, the ANI analysis resulted in identifying individual strains as *Peribacillus frigoritolerans*, *Bacillus licheniformis*, and an unidentified *Ornithinibacillus* species, alongside the novel *Neobacillus* species (179-J1A1-HS strain) being described in this study. This allowed us to characterize two Mars 2020 strains (179-C4-2-HS$^T$ and 179-J1A1-HS) belong to novel *Neobacillus* species.

### Molecular phylogeny and genomic relatedness

The WGS of the above said two Mars 2020 strains were generated during this study. However, comparative ANI analyses of the genomes of Mars 2020 strains with all available genomes in NCBI showed high ANI similarities with the genomes of two other strains (95.1% for V4I25 strain and 96% for the AT2.8 strain). Hence, genomes of these two additional strains were retrieved from NCBI and included in this study for comparative genome characterization. The assembly statistics for all four novel strains are summarized in Table 1. Except for genomes, the strains V4I25 and AT2.8 were not available for this study. The draft assembly of strain 179-C4-2-HS$^T$ exhibited a high N50 value of 4.4 Mb across 14 contigs, indicative of a high-quality assembly. However, strain 179-J1A1-HS had a N50 of 378 kb across 54 contigs, a more fragmented assembly that may require further refinement. The genome sizes of all four strains were approximately 6.2 Mb with the GC content of 38.3%. The total number of predicted genes was 5,832 for 179-C4-2-HS$^T$ and 5,870 for 179-J1A1-HS strains (Table 1). Table 2 presents the similarities among closely related members of the novel species based on ANI, average amino acid index (AAI), digital DNA–DNA hybridization (dDDH), percentage of conserved proteins (PoCPs), and two marker genes (16S rRNA and *gyrB*). The marker genes were also showing >99.7% (16S rRNA gene) and 98.4% (*gyrB* gene) sequence similarities between the Mars 2020 isolates described during this study with the V4I25 and AT2.8 strains. Hence, all four strains that were confirmed as novel species based on the molecular phylogeny (ANI, AAI, dDDH, PoCP, 16S rRNA, and *gyrB* genes) were included in the

**TABLE 1** Genome assembly statistics of various strains belonging to *Neobacillus driksii*

| Strain | References | NCBI Accession no. | Isolation location | Genome size (bp) | No. of contigs | N50 (bp) | No. of filtered reads | Average coverage | G + C content (%) | Coding sequences |
|---|---|---|---|---|---|---|---|---|---|---|
| 179-C4-2-HS[Ta] | This study | JAROBZ000000000 | Clean room floor, JPL Spacecraft Assembly Facility | 6,172,610 | 14 | 4,416,289 | 17,974,640 | 728 | 38.31 | 5,832 |
| 179J 1A1 HS | This study | JBDZYE000000000 | Clean room floor, JPL Spacecraft Assembly Facility | 6,155,344 | 54 | 378,292 | 4,825,790 | 196 | 38.32 | 5,870 |
| AT2.8 | NCBI genome | GCF_013409995.1 | Agave microbiome | 6,500,079 | 31 | 416,293 | 5,980,073 | 230 | 38.35 | 6,256 |
| V4I25 | NCBI genome | GCF_030817595.1 | Lind Dryland Research Station, Washington, USA | 6,973,060 | 3 | 6,733,662 | 920,444 | 33 | 38.42 | 6,605 |

[a]Hybrid assembly using both short-read Illumina and long-read ONT platforms.

genomic comparison, even though V4I25 and AT2.8 strains were not isolated during this study.

The comparative genomic analysis indicated that the four novel strains shared only 86.4% ANI similarities with *N. niacini* NBRC 15566[T] (=DSM 2923[T]) (File S1) and a dDDH value of 32.5% (Table 2) hence these four strains deserve a novel species status. The 16S rRNA gene sequences of Mars 2020 strains, isolated from the JPL-SAF cleanroom surface, the strain AT2.8 isolated from the Agave plant and the strain V4I25 from dryland wheat soil were found to exhibit >99% similarity to *N. niacini* NBRC 15566[T], indicating that the 16S rRNA gene was not a suitable marker for distinguishing members of this genus. Among the four novel strains, the 16S rRNA gene sequence similarity was also >99%. Upon comparison, the *gyrB* sequences showed 90.2% similarity of Mars 2020, Agave, and wheat soil strains with *N. niacini* NBRC 15566[T]. Since it was established that ~95% *gyrB* is the cutoff value for species delineation (9), additional phylogenetic analysis using WGS was also performed, which showed that the ANI, AAI, PoCP indices were only 86.4%, 87.8%, and 77.7%, respectively, between *N. niacini* NBRC 15566[T] and the four novel strains. Based on the low ANI, AAI, PoCP indices and dDDH value (32.5%), the Mars 2020, Agave, and wheat soil strains were differentiated from *N. niacini* NBRC 15566[T] and described as a novel species. The 16S rRNA (Fig. 1A) gene-based phylogeny showed that two Mars 2020 strains, the AT2.8 and V4I25 isolates, and *N. niacini* NBRC 15566[T] formed a tight clade. However, in the *gyrB* phylogeny (Fig. 1B) and the WGS-based tree (Fig. 1C), the four novel strains were clearly separated from all *Neobacillus* species but were found to be closer to *N. niacini* NBRC 15566[T] (ANI 86.4% to 86.8%). The WGS tree was constructed using a concatenated alignment of gene clusters from 26 genomes, and the alignment included a total of 119 single-copy core genes common to all species in the phylum *Bacillota* (formerly *Firmicutes*).

Phylogenetic trees based on the amino acid sequences of two discriminatory proteins, namely imidazole glycerol phosphate synthase subunit [*hisH*] (File S2A) and ATP phosphoribosyltransferase regulatory subunit [*hisZ*] (File S2B), are shown for all reference genomes of the genus *Neobacillus*, including the four genomes of the novel *Neobacillus* species examined during this study. Although *hisH* and *hisZ* trees did not exactly match with the WGS tree, all four novel *Neobacillus* strains clustered together to form a distinct clade, tightly associated with the *N. niacini* type strain (83% for *hisH* and 100% for *hisZ* bootstrap support values).

## Phenotypic and biochemical characteristics

Cells are Gram-staining-positive, motile, strictly aerobic, spore-forming, chemoheterotrophic and mesophilic. Oxidase-positive and catalase-negative. On TSA, after 1–2 days of incubation at 30°C, colonies are circular with regular margins, convex, 0.5–1.0 mm in diameter. Cells are slender rods with rounded edges. Dimensions of the cells are 2.0

**TABLE 2** Sequence similarity percentage of novel strains isolated during this study with all *Neobacillus* species whose genomes are available[a,b]

| WGS accession | Organism name | Strain # | Validity | 16S Accession # | Similarity percentage with respect to *Neobacillus driksii* 179-JC42 HS[T] | | | | | | Similarity with paeninodin gene clusters[c] |
|---|---|---|---|---|---|---|---|---|---|---|---|
| | | | | | 16S % | gyrB % | ANI % | dDDH % | AAI % | PoCP % | |
| JAROBZ000000000000 | *Neobacillus driksii* | 179-C4-2 HS[T] | This study (Type strain) | PP849388 | 100.0 | 99.7 | 97.1 | 74.5 | 96.9 | 89.4 | 80 |
| JBDZYE000000000000 | *Neobacillus driksii* | 179-J1A1 HS | This study | PP849389 | 99.9 | 99.4 | 96.0 | 67.0 | 95.2 | 84.7 | 80 |
| GCF_013409995.1 | *Neobacillus driksii* | AT2.8 | Agave microbiome | ON217559 | 99.7 | 98.4 | 95.1 | 61.8 | 94.5 | 83.7 | 80 |
| GCF_030817595.1 | *Neobacillus driksii* | V4I25 | Lind Dryland Research Station, Washington, USA | N/A | | | | | | | |
| GCF_001591505.1 | *Neobacillus niacini* | NBRC 15566 | Validly published | AB021194 | 99.2 | 90.2 | 86.4 | 32.6 | 87.8 | 77.7 | 80 |
| GCF_001636415.1 | *Neobacillus drentensis* | FJAT-10044 | Validly published | AJ542506 | 99.1 | NS | 79.1 | 21.3 | 73.4 | 65.9 | |
| GCF_030813055.1 | *Neobacillus ginsengisoli* | DSM 27594 | Validly published | HQ224517 | 98.3 | 80.5 | 79.1 | 20.3 | 72.4 | 62.1 | 80 |
| GCA_023702235.1 | *Neobacillus pocheonensis* | KCTC 13943 | Validly published | AB245377 | 98.5 | 80.0 | 79.1 | 21.6 | 72.0 | 59.1 | |
| GCF_001636395.1 | *Neobacillus novalis* | FJAT-14227 | Validly published | AJ542512 | 98.1 | 80.1 | 78.7 | 20.4 | 71.5 | 61.5 | 80 |
| GCF_016908975.1 | *Neobacillus cucumis* | DSM 101566 | Validly published | KT895286 | 98.2 | 79.6 | 78.6 | 20.7 | 71.7 | 66.3 | 100 |
| GCF_002335815.1 | *Neobacillus soli* | DSM 15604 | Validly published | AB681793 | 98.1 | 80.0 | 78.6 | 20.6 | 72.7 | 63.5 | 100 |
| GCF_001026695.1 | *Neobacillus vireti* | DSM 15602 | Validly published | AJ542509 | 98.1 | 80.7 | 78.4 | 19.7 | 71.9 | 61.5 | |
| GCF_937468385.1 | *Neobacillus rhizosphaerae* | CIP 111895 | Validly published | ON543386 | 98.2 | 81.3 | 78.3 | 20.2 | 72.2 | 59.5 | |
| GCF_000307875.1 | *Neobacillus bataviensis* | LMG 21833 | Validly published | AJ542508 | 98.9 | NS | 78.3 | 19.7 | 72.2 | 65.0 | |
| GCF_000612625.1 | *Neobacillus jeddahensis* | JCE | Validly published | HG931339 | 97.4 | NS | 78.3 | 19.6 | 72.7 | 61.9 | |
| GCF_010975035.1 | *Neobacillus thermocopriae* | SgZ-7 | Validly published | JX113681 | 96.5 | NS | 77.8 | 19.0 | 72.5 | 58.1 | |
| GCF_010614825.1 | *Neobacillus sedimentimangrovi* | FJAT-2464 | Validly published | MN963926 | 97.1 | NS | 77.7 | 19.8 | 73.0 | 57.9 | |
| GCF_001591485.1 | *Neobacillus fumarioli* | NBRC 102428 | Validly published | AB681784 | 97.6 | NS | 77.3 | 19.1 | 70.6 | 54.4 | |
| GCF_018343545.2 | *Neobacillus citreus* | FJAT-50051 | Validly published | MZ144021 | 97.9 | NS | <77 | 19.2 | 71.2 | 64.5 | 60 |
| GCF_000612665.1 | *Neobacillus dielmonensis* | FF4 | Validly published | HG315676 | 96.7 | 79.3 | <77 | 19.8 | 70.3 | 62.7 | |
| GCF_013248975.1 | *Neobacillus endophyticus* | BRMEA1 | Validly published | MT501786 | 98.0 | NS | <77 | 21.1 | 69.3 | 55.4 | 80 |
| GCF_014655545.1 | *Neobacillus kokaensis* | LOB 377 | Validly published | LC570960 | 98.0 | NS | <77 | 19.5 | 70.4 | 62.0 | |
| GCF_001048695.1 | *Neobacillus massiliamazoniensis* | LF1 | Not validly published | LK021124 | 98.5 | 79.6 | <77 | 21.7 | 69.9 | 56.5 | |
| GCF_001636315.1 | *Neobacillus mesonae* | FJAT-13985 | Validly published | JX262263 | 97.6 | NS | <77 | 20.1 | 69.7 | 60.9 | |
| GCF_943193175.1 | *Neobacillus muris* | DSM 110989 | Validly published | OM658610 | 98.1 | 77.8 | <77 | 18.8 | 70.1 | 62.1 | |
| GCF_003515685.1 | *Neobacillus notoginsengisoli* | JCM 30743 | validly published | KP076294 | 95.9 | NS | <77 | 22.5 | 64.5 | 53.5 | |
| GCF_016765675.1 | *Neobacillus paridis* | YIM B02564 | Validly published | MW911620 | 97.2 | 78.4 | <77 | 18.9 | 69.0 | 52.4 | |
| GCF_003362805.1 | *Neobacillus piezotolerans* | YLB-04 | Validly published | MG913994 | 95.5 | NS | <77 | 20.0 | 64.7 | 55.1 | |
| GCF_018343535.1 | *Neobacillus rhizophilus* | FJAT-49825 | Validly published | MZ144020 | 98.0 | NS | <77 | 21.0 | 71.1 | 64.4 | |

**TABLE 2** Sequence similarity percentage of novel strains isolated during this study with all *Neobacillus* species whose genomes are available[a,b] (*Continued*)

| WGS accession | Organism name | Strain # | Validity | 16S Accession # | Similarity percentage with respect to *Neobacillus driksii* 179-J C42 HS[T] | | | | | | Similarity with paeninodin gene clusters[c] |
|---|---|---|---|---|---|---|---|---|---|---|---|
| | | | | | 16S % | gyrB % | ANI % | dDDH % | AAI % | PoCP % | |
| GCF_011393025.1 | *Neobacillus terrae* | C11 | Validly published | MN620419 | 96.9 | NS | <77 | 20.5 | 67.5 | 59.0 | 80 |

[a]N/A: Not available but sequence retrieved from the genome.
[b]NS: BLAST results showed it is not significant (<70%).
[c]Percent similarities are calculated based on 100% for the *P. dendritiformis* C454 pathway (4). *Neobacillus* species lacking the paeninodin biosynthetic gene cluster (BGC) are indicated with empty cells.

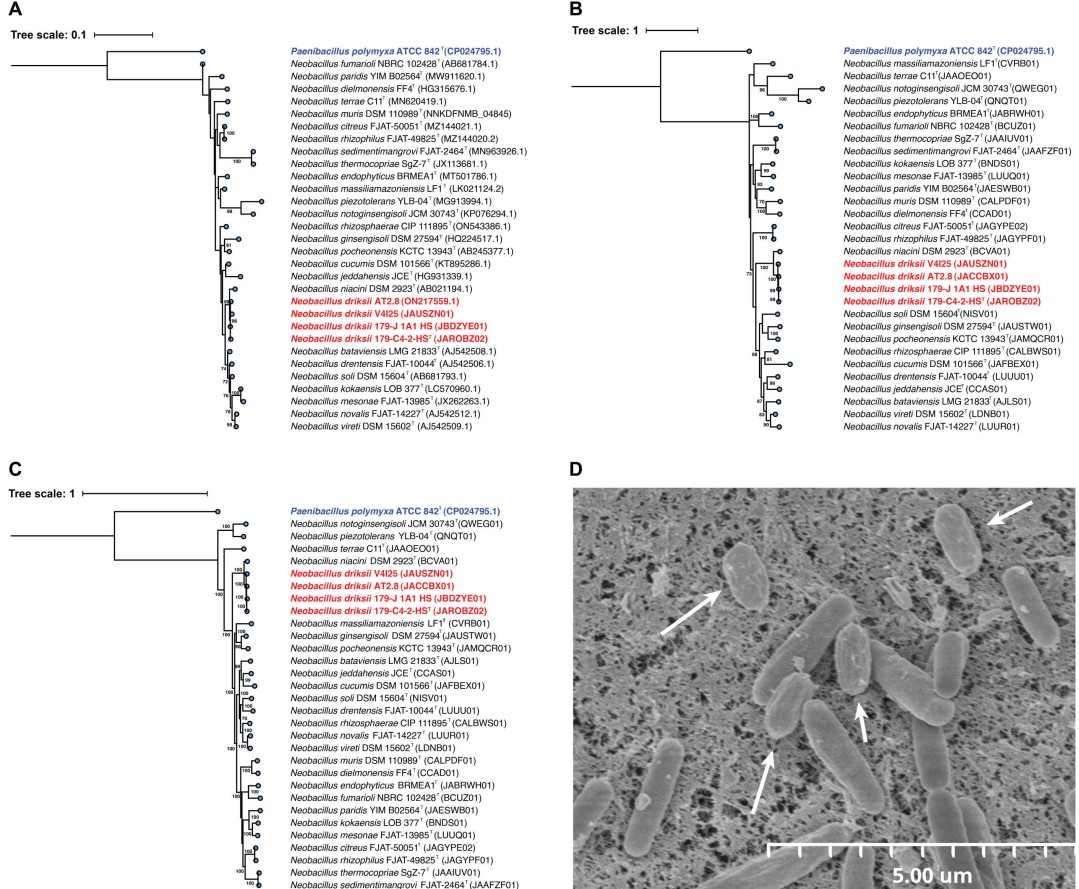

**FIG 1** Molecular phylogeny, genomic relatedness, and morphological features of *Neobacillus driksii* sp. nov. 179-C4-2-HS[T]. (A) Maximum likelihood phylogeny of strains of *N. driksii* (*n* = 4) and 25 established species of *Neobacillus* based on 16S rRNA gene, (B) *gyr*B, and (C) WGS are shown. Corresponding gene sequence from *Paenibacillus polymyxa* ATCC 842[T] was used as an outgroup. Ultrafast bootstrap values >70% are indicated below the branches. (D) Scanning electron microscopy image of *N. driksii* 179-C4-2-HS[T]. Scale bar, 5 µm. Spores are depicted with arrow marks.

± 0.19 µm in length and 0.54 ± 0.06 µm in width, whereas spores are 1.2 ± 0.07 µm in length and 0.56 ± 0.06 µm in width. When strain 179-C4-2-HS[T] was heat-shocked (80°C; 15 min) and regrown in tryptic soy broth (TSB) medium before analyzing for cell morphology, electron microscopy revealed that the cells were marginally thicker and larger in size (data not shown) compared with those from the non-heat-shocked cells (Fig. 1D). Cell growth occurs at 4°C–45°C (35°C optimum), 0.5%–15% NaCl (4–5% optimum) and at pH 6.0–9.5 (optimum, pH 7.5). The biochemical profiles based on BioLog GNIII MicroPlate and Vitek 2 GP are given in File S3.

## Chemotaxonomy

Strain 179-C4-2-HS[T] produced iso-$C_{15:0}$ (43.6%) and anteiso-$C_{15:0}$ (32.7%) as the major cellular fatty acids (File S4) and iso-$C_{14:0}$, $C_{16:0}$ and iso-$C_{16:0}$ in minor quantities, aligning its fatty acid profile with that typical of the genus *Neobacillus*. *Neobacillus* representatives consistently display the predominance of iso-$C_{15:0}$ and anteiso-$C_{15:0}$ in their fatty acid profiles (10). Strain 179-C4-2-HS[T] was also found to produce MK-7 as the major respiratory quinone, consistent with other *Neobacillus* species (10). A1γ with meso 2,6-dimino-pimelic acid is the diagnostic diamino acid of cell wall peptidoglycans for the strain 179-C4-2-HS[T] (File S5).

The strain 179-C4-2-HS[T] produced diphosphatidylglycerol (DPG), phosphatidylglycerol (PG), phosphatidylethanolamine (PE), and an unidentified aminolipid (AL) as polar lipid components. While *Neobacillus* strains can produce several unidentified lipids with

or without glyco-, amino-, phospho-, and aminophospho-moieties (11, 12), the strain 179-C4-2-HS$^T$ contained an unidentified amino lipid, but lacked other polar lipids with glyco-, phospho- and aminophospho-moieties (File S6). The detection of DPG, PG and PE in strain 179-C4-2-HS$^T$ aligns with the polar lipid data reported for other *Neobacillus* species (10).

## Distinguishing characteristics of novel species from other *Neobacillus* species

Based on various microscopy and traditional microbiological analyses, strain 179-C4-2-HS$^T$ and *N. niacini* NBRC 15566$^T$ shared several common morphological, biochemical, and phenotypic features, such as pleomorphic cells, positive for catalase and oxidase activities, anaerobic growth, and acid production from lactose. However, strain 179-C4-2-HS$^T$ can grow at temperature 45°C and pH 6, distinguishing it from *N. niacini* NBRC 15566$^T$. Additionally, strain 179-C4-2-HS$^T$ showed positive gelatin hydrolysis and different acid production patterns compared with *N. niacini* NBRC 15566$^T$. FAME profiles also serve as a discriminatory test; strain 179-C4-2-HS$^T$ produced a higher proportion of iso-C$_{15:0}$ (44%) compared with *N. niacini* NBRC 15566$^T$ (34%), while *N. niacini* NBRC 15566$^T$ had higher synthesis of anteiso-C$_{15:0}$ (41%) than strain 179-C4-2-HS$^T$ (33%). Several phenotypic characteristics (File S3) and FAME profiles (File S4) can be used to differentiate strain 179-C4-2-HS$^T$ from other validly described *Neobacillus* species.

## Description of *Neobacillus driksii* sp. nov.

*Neobacillus driksii* (driks'i.i. *N.L gen*. n. driksii referring to Adam Driks, an accomplished American spore biologist)

Cells are Gram-stain-positive, motile, facultatively anaerobic, chemoheterotrophic, and mesophilic, as well as positive for catalase and oxidase. Cells are non-flagellated, slender rods with rounded ends. Cells are 2.02 µm in length and 0.54 µm in width. Subterminal ellipsoidal spores are found in the swollen sporangia. On TSA, after 1–2 days of incubation at 30°C produces pleomorphic, whitish, irregular colonies having a diameter of 0.5–1.0 mm. Cell growth occurs at 4°C–45°C (35°C optimum), 0.5%–5% NaCl (3% optimum), and at pH 6.0–9.5 (optimum, pH 7.5). It reduces nitrate to N$_2$, produces acid (BioLog GNIII) from D-cellobiose, B-gentiobiose, lactose, melibiose, salicin, starch, sucrose and turanose, and positive reactions (Vitek 2 GP) for L-proline arylamidase, β-glucuronidase, alanine arylamidase, and utilizes D-trehalose. The major (>30%) fatty acids are iso-C$_{15:0}$ and anteiso-C$_{15:0}$. The major polar lipids are DPG, PG, PE, and an unidentified aminolipid. The major isoprenoid quinone is MK-7. A1γ with meso 2,6-diminopimelic acid is the diagnostic diamino acid of cell wall peptidoglycans. The DNA G + C content of the type strain is 38.2 mol% (based on draft genome sequence). As determined by house-keeping gene sequence analysis, and phylogenomics, the species is a new member of the genus *Neobacillus* affiliated to the family *Bacillaceae*. The type strain is 179-C4-2-HS$^T$ (=DSM 115941$^T$ =NRRL B-65665$^T$), isolated from Mars 2020 mission assembly cleanroom floor at JPL-SAF, Pasadena, California, USA.

## Mapping of metagenomics reads generated from the Mars 2020 assembly cleanrooms

The genome of the newly identified *N. driksii* was tracked in metagenomic data sets (236 samples [116 Propidium Monoazide (PMA)-treated and 120 non-PMA-treated sets]) collected from the JPL-SAF over a 6-month period in 2016, during the preparation phase for the Mars 2020 rover components. By analyzing these paired-end shotgun metagenomic data sets and using a threshold of more than 0.5% genome coverage, we detected genomic evidence of *N. driksii* in 17 samples, as shown in Fig. 2. The percentage of aligned assembly length ranged from 0.52% (32 kb) to 0.96% (60 kb) of the 6.17 Mb genome of *N. driksii*. These samples were collected on four different dates, with 13 separate sampling events from March 15 to June 28, 2016. Among the 17 instances where *N. driksii*-specific sequences were identified, three locations (L3, L8, and

L9) were in PMA-treated samples, which are considered to contain intact cells or cells with uncompromised cell walls. However, *N. driksii*-specific sequences were not present in the samples treated with PMA in the locations L1 (179-J1A1-HS) and L4 (179-C4-2-HS[T]) from where two strains of *N. driksii* were isolated. The results suggest that *N. driksii* is present in low relative abundance (dead, viable, and cultivable states) and exhibits a limited spatial–temporal distribution within the SAF cleanroom environment.

## Functional characteristics

Paeninodin, a lasso peptide known for its antimicrobial properties, is one of the BGCs identified in the genomes of all four *N. driksii* strains. Five genes encoding the biosynthesis pathway for lasso peptides were confirmed using two *in silico* tools (antiSMASH and Rodeo). This pathway was not detected in the genome of the closest phylogenetic neighbor, *N. niacini* NBRC 15566[T]. Furthermore, the precursor peptide sequence is only 47% similar between *N. driksii* strains and *P. dendritiformis* C454, while being identical among Mars 2020 strains (Fig. 3A). The paeninodin BGC differs between the studied genomes, with strain AT2.8 lacking nucleotidyl transferase and ABC transporter (Fig. 3B). Putative promoters have been detected, with some sequences occurring in all *N. driksii* strains. In *N. citreus* FJAT-50051, a gene encoding YdcF family protein involved in cell wall biosynthesis is located upstream of lasso peptide BGC. Paeninodin of *P. dendritiformis* C454 is uniquely phosphorylated due to the HPr(Ser) kinase/phosphatase activity. An alignment of novel HPr kinases showed strong sequence similarity between the two Mars 2020 strains, while the other two *N. driksii* strains (AT2.8 and V4I25) and *P. dendritiformis* C454 exhibited only 50%, 67%, and 44% protein sequence similarity, respectively. Despite the sequence divergence, HPr kinase from all *N. driksii* strains showed structural similarity, including HPr kinase from *P. dendritiformis* C454 (TM score ≥0.93), thereby suggesting the similar function. Paeninodin alignment indicated structural differences, with TM score not exceeding 0.47 (Fig. 3C).

 *N. driksii* was differentiated from its closest phylogenetic neighbor, *N. niacini* NBRC 15566[T], by the absence of active paeninodin biosynthetic pathway in *N. niacini*.

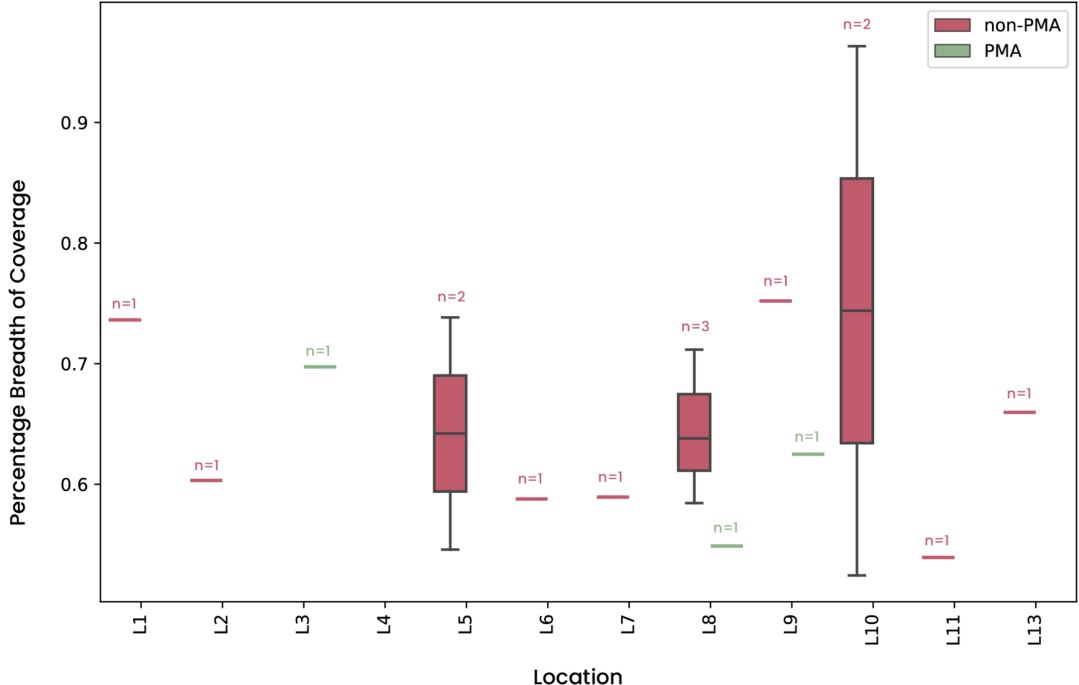

**FIG 2** Mapping of the novel *Neobacillus driksii* to SAF metagenomic reads. Box plots showing the breadth of coverage of the consensus genome were constructed from mapped reads aligned to *N. driksii* 179-C4-2-HS[T] (reported percentage breadth of coverage cutoff >0.5%). The reads from the viable and intact cells (PMA-treated) mapping to the genome of *N. driksii* 179-C4-2-HS[T] were retrieved from three different occasions.

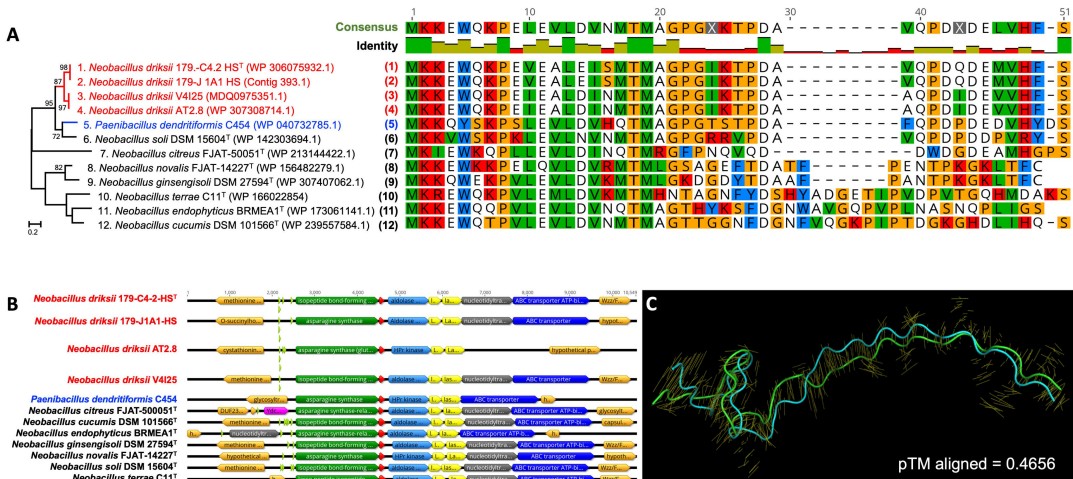

**FIG 3** Molecular characteristics of paeninodin peptide. (A) Maximum likelihood phylogenetic analysis of peptide sequences (bootstrap values > 70% after 1,000 replications are shown in branching point; Scale, 0.1 substitution per position) and alignment of precursor peptides found in *Neobacillus driksii* and other *Neobacillus* species. (B) Overview of the lasso peptide pathway between the paeninodin-producing strains. Asn synthases are marked in dark green, precursor peptides are marked in red, Hpr kinase/phosphatases are marked in light blue, proteins B1 and B2 are marked in yellow, nucleotidyl transferases are marked in gray, and ABC transporters are marked in dark blue. Putative promoters (with Promotech prediction score ≥0.5, see Methods) are marked in light green. Adjacent genes (miscellaneous) are marked in orange and YdcF family protein is marked in pink. (C) Structural alignment of *N. driksii* 179-C4-2-HS[T]/179-J1A1-HS (green) and *N. driksii* AT2.8/V4I25 (cyan) precursor peptide alignment.

Comparative sequence analysis of paeninodin precursor peptides among all *Neobacillus* species (*n* = 26) revealed a heterogeneous distribution of the paeninodin biosynthetic pathway in seven established *Neobacillus* species (Fig. 3A and B; Table 2). The precursor peptides of paeninodin produced by Mars 2020 strains were also found in other environmental *N. driksii* strains, establishing a specific lineage (Fig. 3A). A clade for strains of *N. driksii* was observed, strongly associated (95% confidence of the node) with *P. dendritiformis* C454 and *N. soli* DSM 15604[T], suggesting possible evolutionary relatedness. The protein sequence of paeninodin identified in the novel *N. driksii* described in this study exhibited 80% similarity to the paeninodin found in *P. dendritiformis*. Furthermore, as indicated in Table 1, the biosynthetic pathways of paeninodin in eight *Neobacillus* species, including *N. driksii*, showed similarity percentages ranging from 60% to 100% with those of *P. dendritiformis*. However, additional studies are necessary to determine whether these similarities result from horizontal gene transfer or other evolutionary mechanisms. The data indicated the potential abilities of paeninodin biosynthetic pathways and precursor peptides to discriminate inter-species and intra-species of *N. driksii*.

The novel *N. driksii* strain V4I25, isolated from dryland uncultivated soil next to a wheat field, was identified during a study of phenazine resistance in soil bacteria (8). The V4I25 strain exhibited resistance to phenazine, a natural antibiotic produced by bacteria that has potential for biocontrol of fungal pathogens but can inhibit some bacteria. However, the other three strains of *N. driksii* recognized during this study and other seven *Neobacillus* species should also be tested phenotypically for resistance to phenazine to determine their potential for similar applications.

Additionally, the antiSMASH analysis identified other secondary metabolite clusters, such as the NI-siderophore schizokinen (60% identity) and the betalactone fengycin (40% identity). However, several other potential biosynthetic pathways remain unidentified, indicated by the presence of unknown clusters for type three polyketide synthases (T3PKS), linear azol(in)e-containing peptides (LAP), and terpene (File S7).

The largest enriched group within annotations common for strains 179-C4-2-HS[T] and 179-J1A1-HS is "metabolic functions," where 36 annotations have been found (File S8). When cross-referenced with Cluster of Orthologous Genes (COG) categories, these

annotations include mainly nucleotide metabolism, followed by amino acid, vitamin, and cofactor metabolism as well as energy production and conversion. The most interesting metabolic functions, specific to 179-C4-2-HS[T] and 179-J1A1-HS strains are trehalose synthase (GO:0102986) and P-type proton transporter (GO:0008553). Within the cellular processing/signaling group, 16 annotations were found, including metalloendopeptidase activity (GO:0004222). One annotation indicates information storage and processing class, and one annotation indicates general function only.

A comparative analysis of high-quality score (≥0.5) predicted functions, represented as GO terms, from *N. driksii* genomes has revealed the presence of 73 annotations exclusive to the Mars 2020 strains and 532 annotations shared with strain AT2.8 (Fig. 4A through E). All studied organisms, including strain V4I25 and *N. niacini* NBRC 15566[T], shared 1,118 GO terms. Among the common annotations were functions related to toxin degradation (*e.g.*, azobenzene reductase, GO:0050446), metal ion transport (GO:0030001), and interactions with plant hormones (GO:0010011).

Pairwise clustering of the proteomes revealed the groups of novel sequences, *i.e.,* not present in the other organism (with sequence similarity below 40%, Fig. 4B through 4D). When annotations (GO terms) of these genes were compared, Mars 2020 strains showed a similar number of unique genes with respect to AT2.8 (Fig. 4B) and V4I25 (Fig. 4C), belonging mainly to metabolism or cellular processing and signaling COG categories. Overall, similar unique functions were found in Mars 2020 strains when compared with *N. niacini* genomes (Fig. 4D), with the majority of unique functions belonging to either metabolic or cellular processing and signaling groups (Fig. 4E). Interesting functions, unique only to Mars 2020 strains include xenobiotic transporter (GO:0008559), regulation of small molecule metabolic process (GO:0062012), and steroid dehydrogenase (GO:0016229).

CAZyme prediction in the genomes of all *Neobacillus* strains revealed that *Neobacillus* species divide into two groups *via* k-means clustering, one of which has larger genomes and is significantly more enriched in all types of carbohydrate-active enzymes and which includes *N. driksii* strains. However, the comparatively larger number of CAZymes in this cluster remains true even after each strain's CAZyme counts were normalized by its genome size. *N. driksii* strains cluster tightly together in terms of type and number of

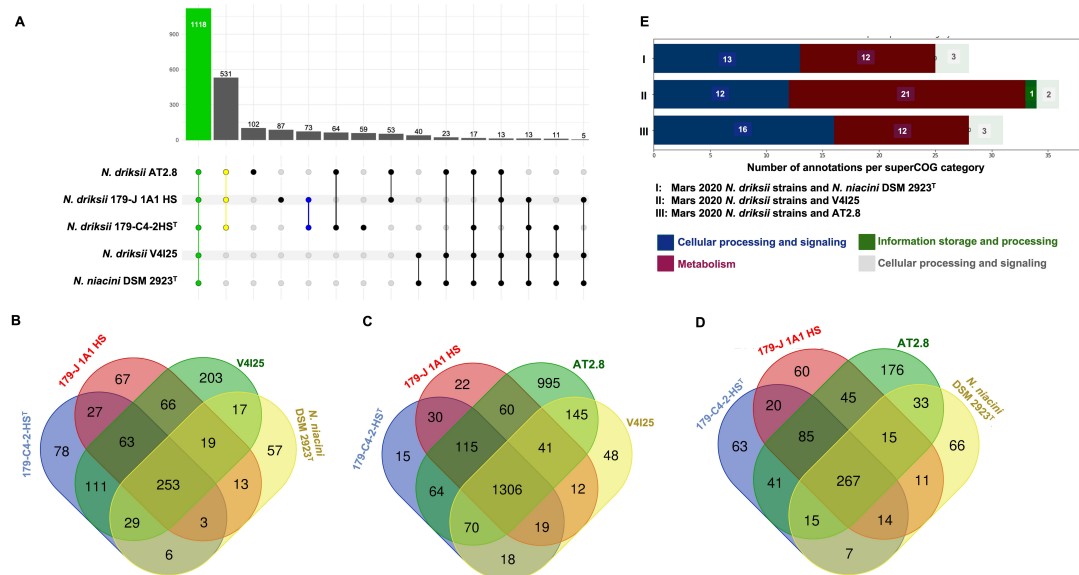

**FIG 4** Comparative functional annotations (HQ GO terms) of strains of *N. driksii* and *N. niacini* NBRC 15566[T]. (A) Entire proteomes; (B) annotations of unique genes derived from CD-HIT-2D clustering against *N. driksii* AT2.8; (C) *N. niacini* NBRC 15566[T]; and (D) *N. driksii* V4I25. (E) Functional clusters of annotations of unique genes shared between *N. driksii* Mars 2020 strains are shown. Digits placed over the bar in A and within Venn's plot (B, C, and D) represent the annotation counts.

CAZymes and are among the strains most enriched in CAZymes in the genus (Fig. 5). Indicator analysis reveals that compared with all other available genomes of *Neobacillus*, *N. driksii* strains are enriched in genes for the degradation of pectins, most specifically rhamnogalacturonan I and II, alginate, and heparin.

## Mobile genetic elements (plasmids)

Genome-wide scan for mobile genetic elements revealed the presence of plasmid-associated genes in all *N. driksii* strains except strain V4l25. In *N. driksii* strains 179-C4-2-HS[T], 179-J1A1-HS, and AT2.8, as well as *N. niacini* NBRC 15566[T], plasmid-associated genes were found to be similar to plasmid pCACC737_3 (NZ_CP059409). However, no genes encoding relaxase or plasmid-related annotations were detected by DeepFRI. Moreover, an additional analysis with Plasmid Finder (v2.1) (13) did not indicate the presence of plasmid in any of the tested genomes. Further analysis with DeepFRI indicated the presence of genes involved in nucleic acid processing, but also genes involved in peptide biosynthesis, metallopeptidases, oxidoreductases (File S8).

## Antimicrobial resistant genes and virulence factors

In total, 43 unique ARGs were identified across the ARG-positive genomes of *Neobacillus* ($n = 81/113$). The mean ARG count per genome was 4.04, range 1 to 11. The most abundant ARG was *rphC* (66 occurrences), followed by *fosM3* (24 occurrences), *fosM1* (20 occurrences), *tetB* (58) (19 occurrences), and *tetA* (58) (15 occurrences) (Fig. 6A). These ARGs are associated with phenotypic resistance to rifampin (*rphC*) *via* antibiotic inactivation, fosfomycin (*fosM1/M3*) also *via* antibiotic inactivation and tetracycline [*tetA/B* (58)] *via* antibiotic efflux.

Across the four *N. driksii* genomes, mean ARG per genome was 3.5, range 1 to 6. Among *N. driksii,* five distinct ARGs were identified to include *rphC* (four occurrences), *fosM2* (three occurrences), *van Y/R (*in *vanF* cluster*)* (two occurrences), *vanS* (in *vanF* cluster)/*vanS* (in *vanM* cluster) (one occurrence), *blt* (one occurrence). The two Mars 2020 *N. driksii* genomes differed by one ARG, given that *fosM2* was identified only in *N. driksii* 179-C4-2-HS[T]. The genome with the highest number of ARGs across *N. driksii* was in V4125 strain, which encoded for six ARGs. Overall, *N. driksii* harbors genetic determinants associated with resistance to rifampin, fosfomycin, fluoroquinolones, and vancomycin. Although disc diffusion susceptibility assays were performed (data not shown), phenotypic interpretation was not possible due to lack of interpretive criteria available for *Neobacillus* species.

In total, 19 unique virulence factors (VFs) were identified across the VF positive genomes of *Neobacillus* ($n = 87/113$). The mean VF count per genome was 5.46, range 1 to 9 (Fig. 6B). In terms of VF genes, *clpP* was universally abundant across the VF positive genomes (87 occurrences) followed by *clpC* and *tufA* with 85 occurrences each. Both *clpP* and *clpC* are associated with stress survival and are chromosomally linked with *Listeria monocytogenes* genomes. The *tufA* gene is chromosomally associated with *Francisella tularensis* and encodes for the microbially abundant elongation factor Tu (EF-Tu). The genome with highest genomic virulence load was that of *N. niacini* PL-B1 ($n = 9$ VFs), originating from JPL 100K clean room (HEPA filter) to include *bspC, bspE, bspF, clpC, clpP, groEL, tutA, wbtE,* and *wbtF*.

Across the four *N. driksii* genomes, mean VF per genome was 6.25, range 6 to 7. Seven unique VF genomic determinants were identified: *bspC/tufA/cplC/cplE/cpl*P (four occurrences), *bspD* (three occurrences), *groEL/wbtE* (one occurrence). The *bpsD* gene was only present across the *N. driksii* genomes, while *groEL* was only present in *N. driksii* 179-J1A1-HS. The *bpsD* is associated with *B. cereus* G9241 plasmid and encodes for *B. cereus* exo-polysaccharide. Apart from the two Mars 2020 *N. driksii*, *bpsD* was also identified in 24 different *Neobacillus* species genomes tested. The *groEL* gene encodes for a chaperonin initially identified in *Clostridium* species chromosomes and has as main function

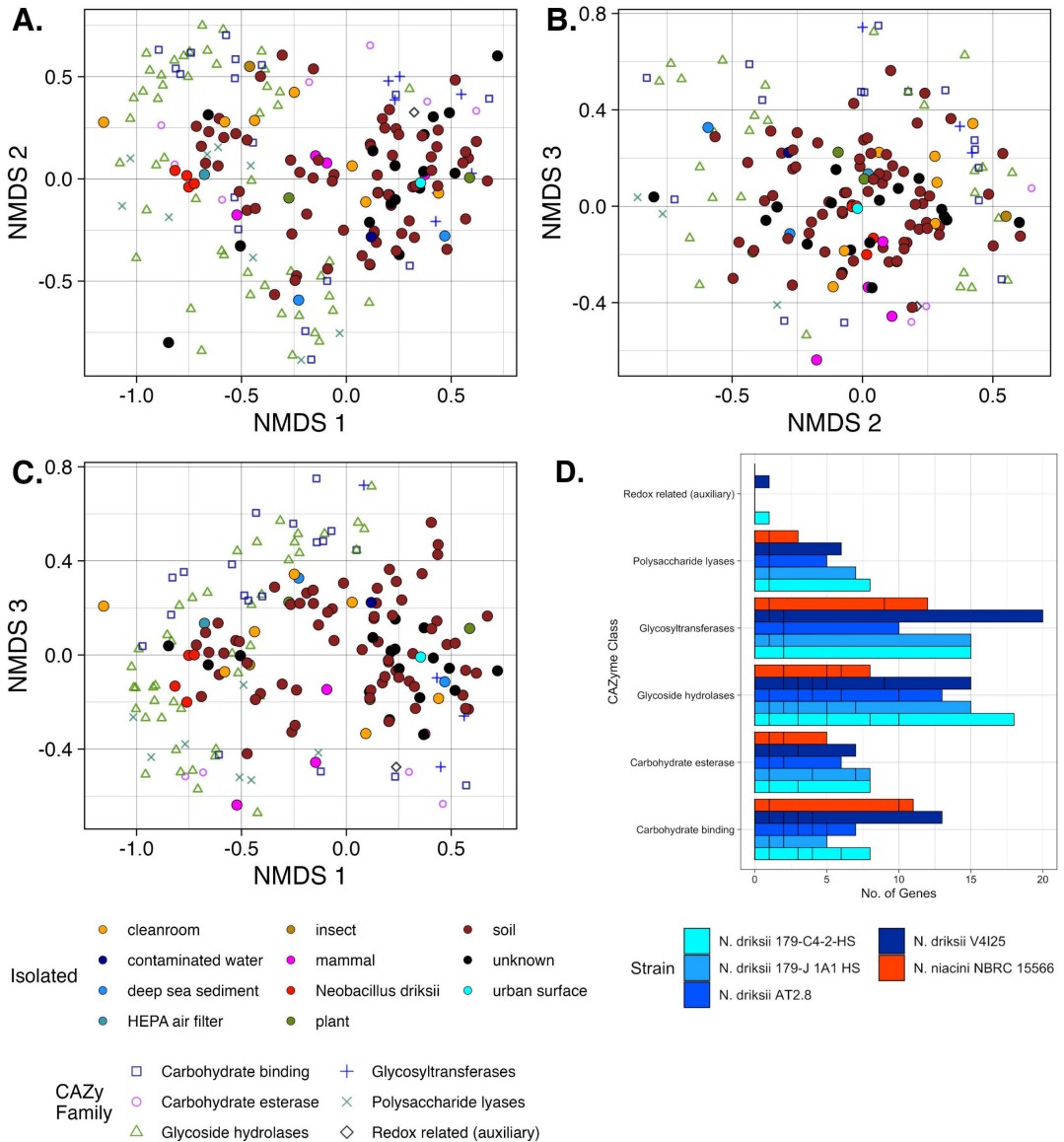

**FIG 5** NMDS ordination of the number of genes encoding CAZymes (family level). (A) Axes 1 and 2, (B) axes 2 and 3, and (C) axes 1 and 3, normalized by genome size of strain for all publicly available strains of *Neobacillus,* as well as strains *N. driksii* 179-C4-2 HS[T] and 179-J1A1-HS Stress: 0.139. (D) Genome size vs sum of CAZyme families detected where each CAZyme count has been normalized by that strain's genome size. *N. driksii* is among the strains with the largest genomes and number of CAZymes (*N. driksii* is part of cluster 2 but is shown as its own cluster for visualization purposes).

responding to heat shock, inducing adaptation to dry and high temperature environments. Subsequently as seen in the Mars 2020 *N. driksii* strains, *groEL* was also identified in 32 different *Neobacillus* genomes tested.

## DISCUSSION

This study provides substantial genomic data highlighting the high similarity between certain novel strains (*n* = 4) within established *Neobacillus* species, alongside their distinctiveness as potentially new species. The high ANI values among the Mars 2020, Agave, and dryland wheat soil strains suggest close genetic relationships, yet phylogenetic analyses indicate clear differentiation from *N. niacini*, underscoring the complexity of microbial classification. ANI values of the *N. driksii* strains below the 95–96% threshold (14, 15), supported by phylogenetic analysis using marker genes (16S rRNA gene and *gyrB*) and WGS-based phylogeny (119 conserved protein markers), confirmed them as

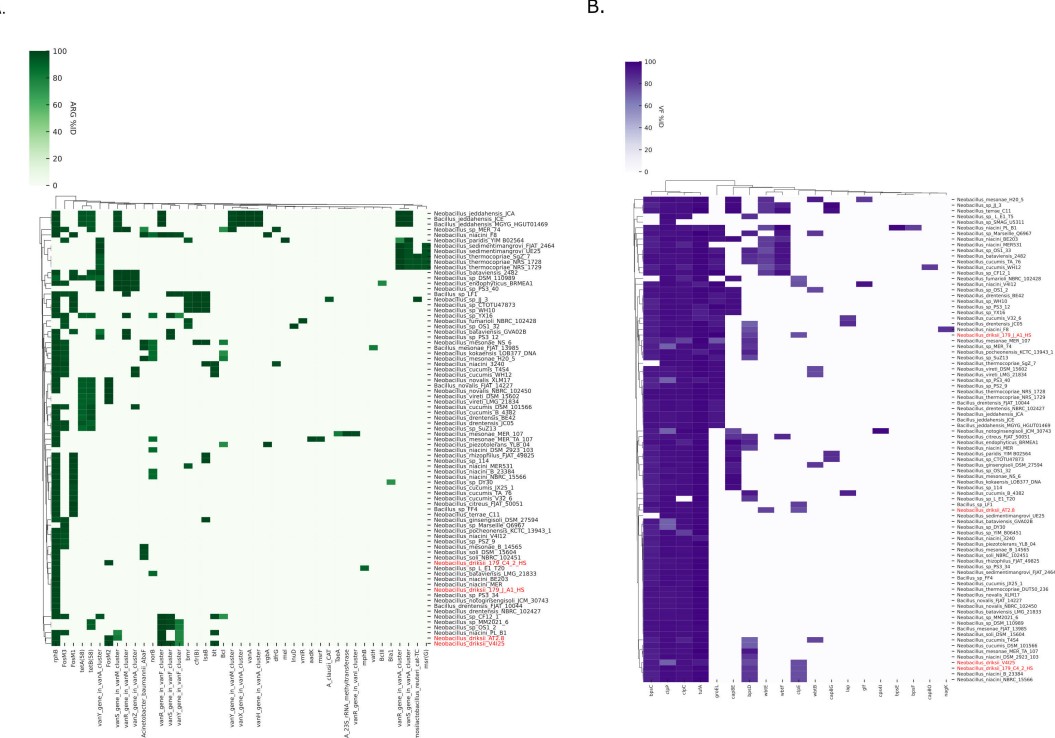

**FIG 6** Genome-wide mapping of antibiotic-resistant genes and genes coding for virulence factors in *Neobacillus driksii* sp. nov. 179-C4-2-HS[T] and other related strains/species. (A) Clustered heat map showing the differential occurrence of ARGs and (B) VFs across all the genomes of the genus *Neobacillus* including four genomes of *N. driksii* (highlighted in red text). Color gradient relates to nucleotide sequence homology between the identified genes and the reference genes in the NCBI-arg and VFDB databases, respectively.

novel species. The dDDH analysis further established the classification of Mars 2020 *N. driksii* isolates as a novel species, with values below the 70% mark indicative of distinct species (16). While genomic data are vital for microbial species delineation, the fundamental characteristics of an organism are often reflected in its phenotypic traits and cell wall composition. Therefore, this research adopted a holistic methodology, integrating genomic findings with phenotypic attributes and chemotaxonomy. With both phenotypic and genotypic evidence obtained adhering to established bacterial nomenclature protocols detailed by the International Code of Nomenclature of Prokaryotes (17), the recognition of a new bacterial species for the Mars 2020 strains was confidently proposed.

Metagenomic analysis in the JPL-SAF cleanroom during the Mars 2020 rover preparation phase revealed the infrequent occurrence of *N. driksii*, identified in only 17 out of 236 samples collected over 6 months in 2016. This intermittent presence suggests that *N. driksii* may have a niche-specific ecology or be a transient organism in the SAF environment. Even though the percentage of cultivable microorganisms among viable populations is extremely low (0.04% to 9%), cleanroom microbiomes are typically dominated by Gram-positive spore-forming bacterial species, comprising 6% to 55% of the cultivable population (18). Previous culture-based analyses in the JPL-SAF have identified 16 bacterial genera, frequently detecting *Bacillus subtilis* and *V. pantothenticus* (19). *Bacillaceae* and "*Amphibacillaceae*" species are notable for their persistence in such environments (20). The resilience of *N. driksii*, detected in non-PMA-treated samples (including dead cells), PMA-treated (viable cells), and traditional plate method (cultivable cells), indicates its ability to survive in various states. This underscores the persistence of certain bacteria despite stringent decontamination processes. Utilizing metagenomic approaches allows for a comprehensive understanding of microbial diversity, as traditional culture-based methods may underestimate

the presence of certain species. However, rare microbes are often not detected in metagenomic sequencing due to insufficient sequencing depth and biases during DNA extraction (21, 22), amplification (23), and sequencing (24). To address this, increasing sequencing depth, using more sensitive detection techniques, and minimizing biases in sample processing are essential.

Lasso peptides are a unique class of ribosomally synthesized and post-translationally modified peptides (RiPPs) characterized by their distinctive lasso-like structure (25). These molecules are renowned for their stability and resistance to proteolytic degradation, making them potent antimicrobial agents. The presence of lasso peptide BGCs in *N. driksii* strains provides strong evidence for these pathways, indicating their potential antimicrobial properties (26). Sequence alignment revealed divergence between Mars 2020 *N. driksii* strains and other *N. driksii* isolates (AT2.8 and V4I25), with each pair being identical (Fig. 3A and B), suggesting evolutionary differentiation within the *Neobacillus* lineage. The absence of paeninodin-related genes in *N. niacini* NBRC 15566$^T$ indicates a gain-of-function divergence, highlighting genetic plasticity. These peptides could offer new avenues for developing novel antibiotics and other therapeutic agents due to their robust structural properties and biological activities (27). Genes encoding paeninodin biosynthesis could serve as taxonomic biomarkers, aiding in the differentiation and classification of *N. driksii* strains from closely related species, thus enhancing our understanding of microbial diversity and evolutionary history.

The presence of secondary metabolites such as NI-siderophore and betalactone in *N. driksii* strains underscores their potential importance in antimicrobial research and biotechnological applications (28). The ability of these strains to produce such potent compounds suggests they could be a valuable resource for developing new antibiotics, antifungal agents, and other therapeutic molecules. NI-siderophores are a type of iron-chelating compound that plays a critical role in microbial iron acquisition. These secondary metabolites are crucial for the survival and virulence of many bacteria, particularly in iron-limited environments. The identification of NI-siderophore biosynthesis pathways in *N. driksii* suggests these strains have adapted to efficiently scavenge iron, a vital nutrient, which could be advantageous for their survival and proliferation in the controlled environments of space missions (28). Betalactones are a class of compounds that contain a β-lactone ring, and they are known for their diverse biological activities, including antimicrobial, anticancer, and enzyme inhibitory properties. The detection of betalactone biosynthesis pathways in the genomes of *N. driksii* strains but not in *N. niacini* indicates their capability to produce these potent secondary metabolites. Betalactones can inhibit serine hydrolases, which are enzymes involved in various physiological processes, making them promising candidates for drug development and biotechnological applications (29).

The functions found in the four *N. driksii* genomes reveal a common origin, as evidenced by shared genes related to plant hormone binding, specifically auxin binding, present in all four genomes (GO:0010011; File S8). Additionally, functional annotation indicates the presence of nitrate reductase genes (GO:0008940), suggesting potential plant interactions across all studied *N. driksii*, despite only one strain being isolated from Agave plants. Environmental factors, such as habitat, influence genomic rearrangements. For instance, the unique metalloendopeptidase (GO:0004222) found in Mars 2020 strains may reflect selective pressure from the cleanroom environment where metals are assembled into spacecraft components. Organisms exposed to increased radiation, such as bacteria from the ISS, have been found to contain additional metallopeptidases (30). Moreover, plasmid NZ_CP059409 contains additional metallopeptidases, as revealed by DeepFRI annotation.

The potential interaction of *N. driksii* with plants and plant tissue is also supported by CAZyme genomic predictions and phenotypic testing. Compared with all other strains of *Neobacillus*, *N. driksii* is comparatively enriched in genes for the degradation of pectin, most specifically rhamnogalacturonan I and II, which are found in plant cell walls, and alginate, a polymer found in brown seaweed with significant industrial applications (31).

This suggests that one of the main ecological roles of *N. driksii* is the degradation of tough plant material. Additionally, one of the two Mars 2020 strains of *N. driksii*, strain V4I25 which was isolated from undisturbed soil next to a wheat field, has been shown to be resistant to phenazine (8, 32). Phenazine is a redox-active molecule produced by various soil bacteria, which has biocontrol properties against fungi and is being tested as a potential biocontrol agent for fungal diseases affecting wheat and other crops; however, there is potential concern that phenazine would also inhibit crucial soil bacteria (8). Because *N. driksii* is enriched in genes for plant cell wall degradation and is resistant to this biocontrol agent (33), this species could be useful for supporting microbial ecological stability (34) and therefore crop health (35) in space agriculture.

The analysis of ARGs in *Neobacillus* genomes identified a relatively narrow diversity of resistance determinants, with 43 unique ARGs identified across 81 ARG positive genomes. Resistance determinants to rifampin (*rphC*), fosfomycin (*fosM1/M3*), and tetracycline [*tetA/B* (58)] were ubiquitously abundant across the genus, a mixture of mechanisms of direct antibiotic inactivation and efflux (36). The relatively conserved ARG range suggests the relevance of these ARGs for *Neobacillus* survival and underscores the importance of characterizing the resistome of environmental microorganisms from an one health perspective as they often serve as reservoirs clinically relevant ARGs in the environmental domain (37). In the specific context of *N. driksii*, the identification of ARGs, such as *rphC*, *fosM2*, and *vanS/Y* across different strains, suggests a genomic capacity for multi-drug resistant (MDR) phenotypes (38). However, this cannot be verified due to lack of relevant susceptibility breakpoints for this genus. The presence of ARGs unique to certain strains, like *fosM2* in *N. driksii* 179-C4-2-HS[T], could reflect strain-specific horizontally acquired ARGs (Fig. 6). As there are currently no clinically relevant breakpoints in use for *Neobacillus* species established by the European Committee on Antimicrobial Susceptibility Testing (EUCAST), it is not possible to fully characterize the phenotypic antimicrobial susceptibility profile of *N. driksii* strain 179-C4-2-HS[T].

The VF analysis further identified a universal presence of stress survival genes like *clpP* and *clpC*, and the frequent occurrence of *tufA* indicates the presence of mechanisms for surviving in stress-induced environments (39). In *N. driksii,* the presence of stress survival VFs such as *clpC* and *groEL* as well as immune evasion associated genes *bsp*, *wbtE* and *tufA* in certain strains, indicates *N. driksii*'s genomic potential in surviving under stress-induced environmental conditions as well as having the genomic potential for human immune system evasion (40).

The resilience and adaptability of microbial communities in extreme and controlled environments, such as NASA mission assembly cleanrooms (resembling hospital intensive care units), highlight their potential for survival and contamination in extraterrestrial ecosystems. This adaptability is crucial for developing contamination control strategies. The isolation of novel strains like *N. driksii* from diverse environments—cleanrooms, Agave plants, and dryland wheat soils—demonstrates their wide ecological adaptability. These microbes employ various survival strategies, including spore formation, heat shock resistance, and specialized metabolic pathways for efficient resource utilization. Furthermore, the genomes of these microbes reveal the presence of several sporulation-related genes (GO:0043934), with varying numbers across strains: nine in SAF strains (179-C4-2-HS[T], 179-J1A1-HS), 10 in V4I25, seven in *N. niacini* NBRC 15566[T], and six in *N. driksii* AT2.8. Their functional properties, such as nucleotide metabolism, energy production, and resistance genes, enable them to thrive in low-nutrient environments and withstand antimicrobial agents and environmental stressors. The production of antimicrobial compounds like lasso peptides provides a competitive advantage. This research enhances our understanding of microbial dynamics in spacecraft assembly facilities and has broad implications for astrobiology, interplanetary travel, and Earth-bound low-biomass environments, informing strategies to mitigate potential contamination in future space missions.

## MATERIALS AND METHODS

### Sample collection and bacterial isolation

The JPL-SAF cleanroom spanned a total surface area of 921.1 m$^2$, with environmental controls ensuring temperature at 20°C ± 4°C and humidity at 30% ± 5%, alongside stringent gowning protocols and weekly sanitation. Although intended to meet ISO-7 standards, the facility was certified to the ISO-8 level. During the sampling at JPL-SAF, the particle count peaked at 8,287 particles/ft$^3$. Particulate collection was performed over a 1 m$^2$ floor section using 9 × 9 in. polyester wipes, following previously established protocols. The collected samples were enclosed in sterile 500 mL glass containers and conveyed to an ISO-7 laboratory for processing. Here, sterile phosphate-buffered saline (PBS; pH 7.4) was introduced (200 mL) and agitated for half a minute to dislodge particulates and microbes. The sample was then concentrated to approximately 5 mL using a 0.45 µm Hollow Fiber Polysulfone-tipped concentrating pipette. To assess the presence of spores, a 425 µL aliquot was subjected to a heat shock at 80°C for 15 min. Another 100 µL was processed without heat treatment to gauge cultivable bacterial populations, adhering to NASA's established guidelines (18). The enumeration of colony-forming units (CFUs) was conducted using the pour plate method with TSA, across four replicates, and colonies were counted following 24, 48, 72 h, and after a week of incubation at 32°C. The strain 179-C4-2-HS$^T$ was isolated on March 15, 2016 and further analyzed for various polyphasic taxonomy tests.

### Physiological tests (temperature, salinity, and pH)

Temperature tolerance and optimum growth temperature were tested by growing the cells in TSB at 4, 10, 20, 27, 35, 40, 50, 70, and 100°C. Salt tolerance and optimum salt concentration required for growth were determined by growing cells in R2A broth supplemented with 0%, 0.5%, 1%, 2%, 3%, 4%, 5%, 6%, 8%, 10%, 12%, 15 and 18% (w/v) NaCl. Tolerance of pH and the optimal pH were determined by growing cells at various pH (4.0–10.0, in intervals of 1.0 pH unit) using the following buffer systems: 0.1 M citric acid/0.1 M trisodium citrate for pH 4.0–5.0; 0.2 M Na$_2$HPO$_4$/0.2 M NaH$_2$PO$_4$ for pH 6.0–8.0 and 0.1 M NaHCO$_3$/0.1 M Na$_2$CO$_3$ for pH 9.0–10.0. pH was checked and adjusted after sterilization of the media.

### Light microscopy and SEM

Strain 179-C4-2-HS$^T$ was cultivated in a liquid medium and subjected to a heat treatment of 80°C for 10 min to facilitate endospore production. Following heat exposure, the culture was transferred onto TSA and incubated at 32°C for a duration of 5 days. For visualization of endospores, the Schaeffer–Fulton staining technique was employed, utilizing malachite green as the primary stain and safranin as the counterstain (41). Microscopic examination was carried out using an Olympus BX53 system, equipped with a DP25 camera, and the images were captured and analyzed with the Olympus cellSens platform. Additionally, Gram staining was executed in accordance with the prescribed protocol using cells grown in TSB for a 24-h period (42).

Strain 179-C4-2-HS$^T$ was grown on nutrient agar at 37°C for 24 h. In addition, to induce spore formation strain 179-C4-2-HS$^T$ was grown in liquid medium and underwent heat shock treatment at 80°C for 10 min. The culture was subsequently grown on nutrient agar at 32°C for 24 h. Squares of 0.5 cm$^2$ consisting of single bacterial colonies were cut out from the agar. The colonies were fixed in EM fixative buffer (2% glutaraldehyde + 2% paraformaldehyde in 0.1 M sodium cacodylate buffer, pH 7.2) for 2 h at room temperature. The fixed samples were stored in 0.2 M sodium cacodylate (pH 7.2) at 4°C. The fixed samples were dehydrated for two 15- min periods in increasing concentrations of ethanol (30%, 50%, 70%, 90%, and 100%). Samples were dried by use of critical point drying protocol (Leica EM CPD300), mounted onto stubs and sputter-coated with gold for 120 s (Quorum Q150R ES Plus). Samples were visualized in SEM (Hitachi S2600N) at

an accelerating voltage of 5.0 kV. Measurements of the length and width of the bacterial cells were made using FiJi software (43).

## Phenotypic and biochemical analyses

Cell motility was tested by placing the cells on TSB supplemented with 0.3, 0.5 and 0.8% agar followed by incubation at 30°C for 48 h. Biochemical tests were carried out on the Vitek 2 GP ID (bioMérieux) card according to the manufacturer's protocol. Freshly grown colonies on TSA were transferred to the saline tube (aqueous 0.45–0.50% NaCl, pH 4.5 to 7.0) until the suspension reached McFarland No. 0.5 as determined using VITEK 2 DensiCHEK Plus. The suspension tube connected to the Vitek 2 GP ID card was placed in the cassette and incubated at 37°C. The sample data entry and retrieval of raw data (within 10 h of inoculation) were done according to the VITEK manual. Acid production was assessed through API 50 CH strip (bioMérieux) according to the manufacturer's instructions. Additionally, GNIII MicroPlate was used to generate a phenotypic fingerprint as per BioLog's protocol. Cells grown on TSA were transferred into the inoculum solution A (Cat #. 72401; BioLog) to reach a McFarland No. 0.50, loaded onto BioLog GNIII (100 µl per well) and incubated at 37°C for 24 h. OmniLog values ($A_{590}$–$A_{750}$) were recorded using the MicroPlate reader (FLUO star Omega, BMG Labtech, Germany) within 24 h of incubation. Antibiotic susceptibility testing (AST) was performed using the Kirby–Bauer disk diffusion protocol (44).

## Analysis of chemotaxonomic parameters

Cells were grown on TSA at 30°C for 48 h until they reached the mid-exponential growth phase. Fatty acid methyl esters (FAMEs) were harvested from cell biomass using sequential saponification, methylation, and extraction (45). Separation of FAME was done using a gas chromatographic instrument (Agilent 7890A) fitted with a flame ionization detector. Microbial Identification System (MIDI) with the Aerobe (RTSBA6) database (Sherlock version 6.0) was used for FAME identification while following the standard protocol (46). Cells were cultivated on TSA for 3 days at 30°C for the extraction of polar lipids, quinones, and peptidoglycans. The polar lipids and quinones were extracted and analyzed by thin-layer chromatography (2D-TLC) (47). Polar lipid spots were identified by spraying the TLC plates with the following reagents: 10% (w/v) ethanolic phospho-molybdic acid for total lipids, 0.2% (w/v) ninhydrin in butanol for aminolipids, Dittmer and Lester's Zinzadze reagent for phospholipids, and alpha-naphthol for glycolipids. The same TLC plate was used for staining amino and phospholipids as per the standard procedure for polar lipid analysis (47). The TLC plate was stained first with ninhydrin reagent to locate aminolipids (pink spots). Subsequently, the same TLC plate was used to stain with Dittmer and Lester's Zinzadze reagent, in which pink spots of amino lipid vanish and blue spots of phospholipids emerge. Total lipids and glycolipids needed additional two TLC plates. Peptidoglycans were extracted and analyzed according to established procedures (48).

## Molecular identification

The Mars 2020 isolates (179-C4-2-HS[T] and 179-J1A1-HS) were cultured on TSA plates at 30°C, and their DNA was extracted using the Mo Bio UltraClean Microbial DNA Isolation Kit (Mo Bio Laboratories, Carlsbad, CA). The 16S rRNA gene segment, approximately 1.5 kb in size, was amplified using established primers (27F; 5′-AGA GTT TGA TCC TGG CTC AG-3′ and 1492R; 5′-GGT TAC CTT GTT ACG ACT T-3′) for further taxonomic identification. The amplification conditions and Sanger sequencing parameters were consistent with previously established methodologies (49). The 16S rRNA gene sequence data were processed with DNASTAR SeqMan Pro software for analysis, and the organisms were taxonomically identified by performing a BLAST against the NCBI 16S rRNA type strain database. Further 16S rRNA gene sequences from all *Neobacillus* were collected and aligned using MAFFT v7.505 with—maxiterate 1,000 and—localpair, and

subsequently phylogenetic trees were derived using the FastTree v.2.1.11 (50). Species delineation was determined by the established sequence similarity threshold of 98.7% (51), and the corresponding 16S rRNA gene sequences were submitted to the GenBank database for archival.

Additionally, the study involved WGS for comprehensive genomic insights. Genomic DNA, isolated using the ZymoBIOMICS DNA MagBead kit, was used to prepare whole-genome sequencing (WGS) libraries with the Illumina Nextera DNA Flex kit. Sequencing was conducted on the NovaSeq 6000 S4 platform, utilizing a paired-end, 2 × 150 base pair configuration. To ensure the high quality of sequencing data, FastQC v.0.11.7 (52) and fastp v.0.20.0 (53) were used to filter out substandard sequences and adapter contamination. The assembly of WGS reads was achieved with SPAdes v.3.11.1 (54). To yield a better resolution genome for the strain 179-C4-2-HS$^T$, Oxford Nanopore sequencing was performed. Strain 179-C4-2-HS$^T$ was revived from cryostock on TSA and genomic DNA was isolated using a Zymo QuikSpin DNA extraction kit. Libraries for Nanopore sequencing were prepared using an SQK-LSK110 Ligation Sequencing kit (Oxford Nanopore) and sequenced on an R9.4.1 flow cell, with base calling *via* MinKNOW/Guppy. Adapter sequences were removed and reads were filtered with the settings "-q 10 -l 200" using Porechop v.0.2.4 (55) and Nanofilt v.2.3.0 (56). Hybrid genome assembly for 179-C4-2-HS$^T$ was prepared by Unicycler v.0.5.0 (57) using Nanopore reads to polish the Illumina assembly. Assembled genome quality was stringently checked with QUAST v.5.0.2 (58) and CheckM v.1.2.2 (59), ensuring over 99% completion and minimal contamination.

Species verification was done through fastANI v.1.34 analysis (60), confirming identities with an ANI of over 95%. Additionally, dDDH estimates were generated with the Genome-to-Genome Distance Calculator (GGDC) v.3.0, employing its recommended Formula 2 alongside the BLAST+ alignment tool (61). We further calculated AAI (62) and PoCP (63) with all the related type species.

The phylogenetic positioning of the new bacterial strains within its genus was determined using GToTree v.1.8.2, a command-line tool using Hidden Markov Models (HMM) to identify and align single-copy genes from respective genomes (64). This is a crucial process that relies on the unique nature of single-copy genes to mitigate the confounding effects of gene duplication and horizontal gene transfer, thus refining evolutionary insights. By using a unique set of 119 single-copy genes for *Bacillota*—these sequences were aligned and concatenated, forming the basis for constructing phylogenetic trees that offer a comprehensive view of evolutionary relationships. *Paenibacillus polymyxa* ATCC 842$^T$ was selected as an outgroup for the phylogenetic trees to provide evolutionary context. Phylogenetic trees were reconstructed using IQ-TREE v.2.2.0.3 (65), leveraging ModelFinder-Plus (66) to select the most suitable evolutionary models with 1,000 ultrafast bootstrap replicates to ensure statistical robustness. The trees were further annotated and visualized in iTOL v.6.0 (67), enabling understanding of the strains' positions within its genus.

## Multisequence alignment and phylogenetic inference

Amino acid sequences for two distinct proteins (ATP phosphoribosyltransferase regulatory subunit [*his*Z] and imidazole glycerol phosphate synthase subunit [*his*H]) were extracted from a total of 29 genomes including all reference genomes of type species of the genus *Neobacillus*, the genomes of four strains of novel species described in this study and the reference genome for *Priestia aryabhattai* K13$^T$ which was used as an outgroup. Each protein's sequences were aligned using the MAFFT v.7.505 (68). The alignments were performed with the options—maxiterate 1,000 and—localpair, to include 1,000 iterative refinements and use the Smith–Waterman algorithm for accurate local pairwise alignments.

Post-alignment, aligned sequences of each protein were concatenated in a consistent order across all genomes to form a comprehensive supermatrix for subsequent phylogenetic analyses. The concatenated supermatrix was analyzed using RAxML v.7.2.8

(69), employing the maximum likelihood method. RAxML analysis utilized automatic protein model selection with gamma-distributed rate heterogeneity (-m PROTGAM-MAAUTO), specified random seeds for reproducibility (-p 12345 and -x 12345), 1,000 bootstrap replicates (-# 1000), and rapid bootstrapping combined with maximum likelihood search (-f a).

## Mapping of cleanroom facility metagenomes to measure the abundance and prevalence of the novel bacterial species

The shotgun metagenome reads were not generated during this study but downloaded from the NCBI (PRJNA1150505). In this study, we focused on identifying new microbial species in regulated environments and also mapped its presence and persistence by analyzing 236 paired-end shotgun metagenomic samples from the JPL-SAF that contained reads associated with PMA treated and untreated samples. When samples were treated with PMA, the dye first intercalates with DNA subsequently on photo-activation, covalently binds DNA, and hence DNA not amplified during PCR process and related molecular analyses. Since this PMA treatment steps eliminates free DNA and also penetrates the cell wall of the compromised microbial cells, the resulting metagenome reads were associated with intact and viable microorganisms (19, 70, 71). Initial quality control of the metagenomic data of both PMA-treated and untreated samples was performed using fastp v.0.22.0 (53), which removed low-quality sequences and other potential artifacts. Following this, we used MetaCompass v.2.0 to map the cleaned reads against the newly sequenced genomes (72). This step was essential for constructing consensus sequences from the mapped read, enabling us to not only quantify the read abundance for each novel species but also assess how well the consensus sequences represented the actual diversity in each sample.

## Genome annotation, resistance, and virulence genomic determinant characterization

Initially, we applied the command-line tool Prokka v.1.14.5 with the—compliant extension (73) to annotate the genomes and detect open reading frames (ORFs). Prokka integrates the gene prediction algorithm Prodigal and cross-references multiple curated databases to annotate the genetic elements accurately. This comprehensive gene annotation process is foundational, as it provides a detailed catalog of putative genes and their predicted functions within the novel bacterial genomes (73).

Once the genomes were annotated, we proceeded with the functional classification of the genes using cogclassifier v.1.0.5, a Python-based tool available at the Python Package Index. This tool mapped the annotated genes to the Clusters of Orthologous Genes (COGs) database, which is a collection of proteins or genes that are orthologous across multiple species. By assigning genes to specific COGs, we were able to construct functional profiles for each of the novel strains, giving us valuable insights into their potential biological roles and evolutionary relationships. An essential part of the analysis focused on the identification of genes associated with antibiotic resistance, a rising concern in global health. Antibiotic resistance genes (ARGs) and VF-encoding genes were classified using Abricate v. 1.0.0 against NCBI and VFDB (74) databases, considering only hits with >70% nucleotide coverage and >70% identity.

## Screening of secondary-metabolite biosynthetic potential

Our analysis also delved into exploring secondary metabolite Biosynthetic Gene Clusters (BGCs) within the genomes. These clusters indicate the ability of microorganisms to produce valuable secondary metabolites, which include antibiotics, pigments, and toxins often with pharmaceutical and biotechnological significance. To conduct this analysis, we utilized antiSMASH v.7.0.0 (75), a tool for BGC detection and analysis, employing a "Strict" detection setting to ensure high specificity in the search. Identified BGCs underwent further refinement through functional annotation against the Minimum Information about a Biosynthetic Gene cluster (MIBiG) database v.3.1 (76). In addition

to antiSMASH analysis, precursor peptides have been characterized using the RODEO tool that specifically detects lasso peptides. Additionally, lasso peptide BGCs have been analyzed for the presence of promoter sequences using Promotech (77), using RF-TETRA and RF-HOT models. Sequences upstream of the clusters (40 nt fragments) with prediction score ≥0.5 were highlighted in Fig. 3B.

## Functional characterization

To identify the functions of protein-coding genes in bacterial genomes, we first detected open reading frames (ORFs) using Prodigal (https://github.com/hyattpd/Prodigal). Subsequently, we translated the gene sequences to amino acids and annotated them using a metagenomic pipeline (https://github.com/bioinf-mcb/Metagenomic-DeepFRI/) incorporating the DeepFRI tool (78). This pipeline generated a query contact map by utilizing results from the mmseqs2 target database (https://github.com/soedin-glab/MMseqs2) to find similar protein sequences with known structures. We used the contact map alignment as input for the DeepFRI Graph Convolutional Network (GCN), and processed annotations without known structure matches using a Convolutional Neural Network (CNN). In this study, we employed DeepFRI v.1.1, retraining the original DeepFRI architecture with high-quality models from the AlphaFold Database, which enhanced the number of functions the method could predict by up to four times (79). Furthermore, we predicted the structures of HPr serine kinase/phosphatase and precursor peptide using the AlphaFold database (79, 80) and assessed structural alignment using template modeling (TM) score (81). We identified carbohydrate-active enzymes (CAZymes) in the genomes of all *Neobacillus* species using the dbcan3 pipeline (82). Additionally, we scanned all contigs generated for each genome for potential plasmids using PLASme software (https://github.com/HubertTang/PLASMe), mob-recon of mob-suite v3.1.9. and Plasmid-Finder-2.1 (13).

## ACKNOWLEDGMENTS

We would like to thank Zymo Research Corp. for extracting DNA and Chris Mason, Weill-Cornell Medicine for generating shotgun sequencing. We acknowledged the Jet Propulsion Laboratory supercomputing facility staff, notably Narendra J. Patel (Jimmy) and Edward Villanueva, for their continuous support in providing the best possible infrastructure for BIGDATA analysis. The authors would like to acknowledge the facilities and the scientific and technical assistance of the Anatomy Imaging and Microscopy Facility at the University of Galway (https://imaging.universityofgalway.ie/imaging/). We thank Tomasz Kosciolek from Sano Centre for Computational Personalised Medicine, Krakow, Poland for guiding LS in the paeninodin structural alignment analysis and functional annotations. © 2024 California Institute of Technology, Government sponsorship acknowledged.

Part of the research described in this publication was carried out at the Jet Propulsion Laboratory, California Institute of Technology, under a contract with National Aeronautics and Space Administration. This research was funded by the JPL Mars Program award to KV in 2016. PS is a recipient of the Prime Minister's Research Fellowship from the Ministry of Education, Government of India. SKMS acknowledges the half-time teaching assistantship from the Ministry of Education, Government of India. KR acknowledges support from the Centre for Integrative Biology and Systems mEdicine (IBSE) and the Wadhwani School of Data Science and AI (WSAI), Indian Institute of Technology Madras. The funders had no role in study design, data collection and interpretation, the writing of the manuscript, or the decision to submit the work for publication.

K.V. managed the strain collection, its genome sequencing project, and conceived and designed the study. A.H., F.M., and P.D.R. performed the microbiological experiments and carried out the phenotypic assays. A.H. and P.D.R. performed Vitek and BioLog based biochemical characterization; F.M. and G.M. generated antimicrobial susceptibility testing. K.V. performed light-microscopy assays; F.M. and G.M. generated SEM. images. A.C.S. performed initial phenotypic characterization of the strain and along with N.S.K.

analyzed shotgun metagenome sequences of Illumina platform and *de novo* assembly. P.S., S.K.M.S., and K.R. performed genomic and metagenomic characterization; and L.S. executed the functional annotation. A.H. generated the first draft and K.V. edited the manuscript. All authors read and approved the final manuscript.

## AUTHOR AFFILIATIONS

[1]Division of Microbiology and Biotechnology, Yenepoya Research Centre, Yenepoya (Deemed to be University), Mangalore, India

[2]Antimicrobial Resistance and Microbial Ecology Group, School of Medicine, University of Galway, Galway, Ireland

[3]Department of Biotechnology, Bhupat and Jyoti Mehta School of Biosciences, Indian Institute of Technology Madras, Chennai, India

[4]Center for Integrative Biology and Systems mEdicine (IBSE), Indian Institute of Technology Madras, Chennai, India

[5]Wadhwani School of Data Science and AI, Indian Institute of Technology Madras, Chennai, India

[6]Centre for One Health, Ryan Institute, University of Galway, Galway, Ireland

[7]Sano Centre for Computational Personalised Medicine, Krakow, Poland

[8]Biotechnology and Planetary Protection Group, Jet Propulsion Laboratory, California Institute of Technology, Pasadena, California, USA

[9]Blue Marble Space Institute of Science, Seattle, Washington, USA

[10]Department of Data Science and AI, Wadhwani School of Data Science and AI, Indian Institute of Technology Madras, Chennai, India

## AUTHOR ORCIDs

Asif Hameed http://orcid.org/0000-0003-4445-7080

Francesca McDonagh http://orcid.org/0000-0002-0830-4899

Pratyay Sengupta http://orcid.org/0000-0002-0184-9335

Georgios Miliotis http://orcid.org/0000-0002-0944-2206

Shobhan Karthick Muthamilselvi Sivabalan http://orcid.org/0000-0002-5452-9714

Lukasz Szydlowski http://orcid.org/0000-0003-3953-8417

Anna Simpson http://orcid.org/0000-0002-9493-2030

Nitin Kumar Singh http://orcid.org/0000-0001-5344-1190

Punchappady Devasya Rekha http://orcid.org/0000-0002-9187-6395

Karthik Raman http://orcid.org/0000-0002-9311-7093

Kasthuri Venkateswaran http://orcid.org/0000-0002-6742-0873

## FUNDING

| Funder | Grant(s) | Author(s) |
| --- | --- | --- |
| NASA \| Jet Propulsion Laboratory (JPL) | Mars Program Office | Kasthuri Venkateswaran |
| Indian Institute of Technology Madras (IITM) | IBSE | Karthik Raman |
| Indian Institute of Technology Madras (IITM) | WSAI | Karthik Raman |

## AUTHOR CONTRIBUTIONS

Asif Hameed, Formal analysis, Methodology, Writing – original draft | Francesca McDonagh, Formal analysis, Methodology, Visualization | Pratyay Sengupta, Data curation, Formal analysis, Methodology, Software, Visualization | Georgios Miliotis, Formal analysis, Methodology, Software, Visualization | Shobhan Karthick Muthamilselvi

Sivabalan, Data curation, Formal analysis, Software, Visualization | Lukasz Szydlowski, Data curation, Formal analysis, Methodology, Visualization | Anna Simpson, Data curation, Formal analysis, Methodology, Visualization | Nitin Kumar Singh, Data curation, Formal analysis, Investigation, Visualization | Punchappady Devasya Rekha, Formal analysis, Methodology, Resources | Karthik Raman, Formal analysis, Funding acquisition, Resources, Supervision, Visualization.

## DATA AVAILABILITY

The genomic data for the bacterial isolates studied, including the 16S rRNA gene (strain 179-C4-2 HS[T]: PP849388; 179-J1A1-HS: PP849389) and WGS (strain 179-C4-2 HS[T]: JAROBZ000000000.2; 179-J1A1-HS: JBDZYE000000000.1), have been submitted to the NCBI and are part of BioProject PRJNA935338. Both the Illumina and ONT reads are made available under the given accessions. Illumina reads for strain 179-C4-2 HS[T]: SRR29255618; 179-J1A1-HS: SRR29255617; ONT reads for 179-C4-2-HS[T]: SRR30855205. Accession numbers for these sequences are listed in Table 1. These initial sequences are the ones referenced in the database. The scripts are available on GitHub at https://github.com/RamanLab/Neobacillus/wiki.

## ADDITIONAL FILES

The following material is available online.

### Supplemental Material

**Supplemental material (Spectrum01376-24-s0001.pdf).** Supplemental figures and tables.

### Open Peer Review

**PEER REVIEW HISTORY (review-history.pdf).** An accounting of the reviewer comments and feedback.

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
