## [Reviewer comments · Microbiology Spectrum]

Microbiology Spectrum

Neobacillus driksii* sp. nov. isolated from a Mars 2020 spacecraft assembly facility and genomic potential for lasso peptide production in *Neobacillus

Asif Hameed, Francesca McDonagh, Pratyay Sengupta, Georgios Miliotis, Shobhan Karthick Sivabalan, Lukas Szydowski, Anna Simpson, Nitin Singh, Punchappady Rekha, Karthik Raman, and Kasthuri Venkateswaran

Corresponding Author(s): Kasthuri Venkateswaran, California Institute of Technology

Review Timeline:

Submission Date:	June 19, 2024
Editorial Decision:	August 5, 2024
Revision Received:	August 29, 2024
Editorial Decision:	September 16, 2024
Revision Received:	October 2, 2024
Accepted:	November 5, 2024

Editor: Gaurav Sharma

Reviewer(s): Disclosure of reviewer identity is with reference to reviewer comments included in decision letter(s). The following individuals involved in review of your submission have agreed to reveal their identity: Srikrishna Subramanian (Reviewer #1)

Transaction Report:

DOI: <https://doi.org/10.1128/spectrum.01376-24>

Re: Spectrum01376-24 (*Neobacillus driksii* sp. nov. isolated from a Mars 2020 spacecraft assembly facility and genomic potential for lasso peptide production in *Neobacillus*)

Dear Dr. Kasthuri Venkateswaran,

Thank you for submitting your manuscript to Microbiology Spectrum. We have received comments from two reviewers regarding your submitted manuscript. Both reviewers have recommended additional experiments, explanations, and significant concerns. Therefore, I invite you to revise your manuscript in light of the referees' comments. Your manuscript needs major revisions, and the referees' remarks are appended below for your attention. Please ensure that the added explanations and comments in the rebuttal letter are well explained throughout the text and supported by clearly described methods.

Please return the manuscript within 60 days; if you cannot complete the modification within this period, please get in touch with me. If you do not wish to modify the manuscript and prefer to submit it to another journal, notify me immediately so Spectrum may formally withdraw the manuscript.

Revision Guidelines

To submit your modified manuscript, log into the submission site at <https://spectrum.msubmit.net/cgi-bin/main.plex>. Go to Author Tasks and click the appropriate manuscript title to begin. The information you entered when submitting the paper will be displayed; update this as necessary. Note the following requirements:

Sincerely,
Gaurav Sharma, Ph.D.
Editor, Microbiology Spectrum

Assistant Professor, IIT Hyderabad, India
Lab Page: <https://sites.google.com/view/sharmaglab/>

Reviewer #1 (Comments for the Author):

In this manuscript, the authors report the characterization of two novel microbes isolated from a spacecraft assembly facility via polyphasic taxonomy, Illumina genome sequencing, and draft genome assembly, followed by phylogenomics analysis. The authors report the presence of genes that can help synthesize lasso peptides by the microbes that share a high similarity with those reported here. Gene clusters that likely code for siderophores that could play a role in sequestering heavy metals from contaminated environments were also identified in these microbes. This study is interesting from the perspective of how such microbes may survive in these clean room environments. The analysis is very well defined on the wiki: <https://github.com/RamanLab/Neobacillus/wiki>. While the manuscript covers all essential aspects, I feel some points could be addressed better, as listed below:

Data availability: The metagenomics data reported in the manuscript don't seem to be available. Likewise, genomes referred to in line 139 of page 4 are also unavailable.

Page 3 lines 92-93: Here it is claimed that the two strains did not match with any known bacterial species with ANI >87%. Later in the manuscript, similarity to a couple of other strains described in Table 1 and SF1 with ANI values > 95% is mentioned. Is this because these genomes were not available in NCBI at the time of publication of ref 3?

Page 3 line 107: The logic of sequencing samples with and without treatment with propidium monoazide could be explained

Page 4, line 139: Results section: Here, it is mentioned that WGS was carried out for 17 + 13 isolates. If this work is part of this manuscript, please provide a link to the data and mention how the isolates were characterized. Was it based on 16S or ANI? Likewise, what method was employed for the characterization of the four uncharacterized strains? Did the authors also look at AAI and PoCP in addition to ANI?

Page 4 line 153-161: The section on molecular phylogeny and genomic relatedness is very confusing and poorly written. Table 1 lists four novel strains; however, only two of these were sequenced by this group. Also, the statistics 14 contigs and 54 scaffolds. Is this correct? What does constructed with high-quality sequences mean?

Page 4 line 163: claims that the four strains share considerable genomic similarity with *N. niacini* with ANI of 86.4%. Yet, in the introduction similarities <87% (see line 93) don't mean considerable genomic similarity.

Page 5 lines 184-187: The rationale for selecting *hisH* and *hisZ* is not provided. Also, the claim made in line 190 doesn't appear to be correct based on what is shown in the supplementary figures 2A and 2B.

Another general comment: Since the two genomes reported here are draft in multiple contigs, a tool like contigtax could be run to ascertain no contamination.

Page 6 line 253: Data from the metagenomics sequencing are not available. Please provide links to the same.

Page 7 line 299: Are you suggesting horizontal gene transfer here?

Page 8 line 350: Have you looked for plasmid genes in the two smaller contigs of V4I25? Also, given that the genomes reported here are draft genomes any contigs that are likely to be from plasmids could be highlighted.

Page 9 line 388: There is an unnecessary *niaci* before *Apart*

Page 10 lines 441-443: Genes encoding paeniodin biosynthesis are suggested to serve as taxonomic biomarkers. However, from the analysis this claim is not justified. Please elaborate as to why these are good taxonomic biomarkers.

Page 10 lines 460-461: Appears to be a very loose statement. Also, no discussion about the plant hormone binding genes are mentioned elsewhere in the manuscript.

Page 11 lines 467-469: It may be worthwhile to map the reads from the various genomes to this plasmid and show how much of it is present since plasmids can vary widely in gene content even among closely related microbes.

Page 11 line 512: some analysis of sporulation genes in these microbes may be presented.

Page 14 lines 615-627: How much data was obtained? Data coverage and genome assembly statistics could be provided.

Page 14: In line 635, the outgroup is *Paenibacillus polymyxa*, and in line 646, the outgroup is *Priestia aryabhatai*. Why were different outgroups chosen?

Page 16 line 722: The wiki seems to suggest that the query was *Agrococcus*. Is this correct?

```
# esearch to get the list of representative species accession IDs
esearch -query '[ORGN] AND "representative genome"[filter] AND all[filter] NOT
anomalous[filter]' -db assembly | esummary | xtract -pattern DocumentSummary -def "NA" -element
```

AssemblyAccession > _repr.txt

```
# bit to fetch the genomes  
bit-dl-ncbi-assemblies -w _repr.txt -f fasta -j 100
```

Reviewer #2 (Comments for the Author):

This manuscript reports detailed characterization of two novel strains of Gram-positive bacteria (179-C4-2-HS, 179-J1A1-HS), isolated from the Mars 2020 spacecraft assembly facility clean room, which survived heat treatment (80°C; 15 min). The results from ANI and other genomic similarity analyses show that these two strains are closely related (i.e. within the accepted species range) to the sequences of two other strains (AT2.8 from the Agave plant, and V4I25 from wheat-associated soil), whose genomes were available in the public database. This manuscript describes characterization of these four strains, based on their genome sequences, using multiple approaches. In phylogenetic trees, these strains clustered together, and they were most closely related to the species *Neobacillus niacini* NBRC 15566T. The 16S rRNA gene sequence similarity (>99.2%), average nucleotide identity (ANI) value (86.4%), *gyrB* sequence similarity (90.2%), and digital DNA-DNA hybridization values (32.5%) also supported the inference that these strains differ significantly from their closest relative *N. niacini* NBRC 15566T, and thus constitutes a novel species named *N. driksii*, within the genus *Neobacillus*.

The authors have characterized this new species regarding its phenotypic, biochemical and chemotaxonomic characteristics and have identified some properties which distinguish this species from *N. niacini* NBRC 15566T and other *Neobacillus* species. One novel aspect of *N. driksii* and 7 other *Neobacillus* species reported here is that their genomes encode the biosynthesis pathway for lasso peptides, which possess antimicrobial properties. The authors also report metagenomic analysis showing that *N. driksii* is not commonly found in NASA cleanrooms.

Comments:

The main inference from this study that the strain 179-C4-2-HS isolated from SAF clean room represents a novel species within the genus *Neobacillus*, showing close relationships to *N. niacini* NBRC 15566T, is strongly supported by the presented data. Similarly, the inference that the genomes of *N. driksii* and 7 other *Neobacillus* species contain the genes encoding the biosynthesis pathway for lasso peptides is also supported by the presented analyses. Overall, this is a detailed and presented work and I have no significant comments on it.

Minor comments:

1. In the etymology of *N. driksii*, the initial part on line 234 should be corrected as follows: *Neobacillus driksii* (*driksii*.i. N.L. gen. n. *driksii*

2. As the authors have noted in the Introduction, the genus *Neobacillus* was circumscribed based on the shared presence of several conserved indels in protein sequences. As *N. driksii* is a new species in this genus, these distinguishing indels in this species should be examined. This can be easily done using the [AppIndels.com](https://appindels.com/) server. <https://appindels.com/>

3 I have noted a few typos on Lines 740 (half instead of halt) and 742 (Medicine instead of mEdicine). The manuscript should be thoroughly checked in this regard.

Re: Spectrum01376-24 (*Neobacillus driksii* sp. nov. isolated from a Mars 2020 spacecraft assembly facility and genomic potential for lasso peptide production in *Neobacillus*)

Dear Dr. Kasthuri Venkateswaran,

Thank you for submitting your manuscript to Microbiology Spectrum. We have received comments from two reviewers regarding your submitted manuscript. Both reviewers have recommended additional experiments, explanations, and significant concerns. Therefore, I invite you to revise your manuscript in light of the referees' comments. Your manuscript needs major revisions, and the referees' remarks are appended below for your attention. Please ensure that the added explanations and comments in the rebuttal letter are well explained throughout the text and supported by clearly described methods.

Ans: The editor's and reviewers' comments were instrumental in refining and clarifying the manuscript. We have carefully addressed each of the critical points raised and have made the necessary modifications to improve the overall quality of the manuscript. Below is an itemized list of our responses to the reviewers' comments, reflecting the changes made in the revised version, which is now resubmitted for further consideration.

Reviewer #1 (Comments for the Author):

In this manuscript, the authors report the characterization of two novel microbes isolated from a spacecraft assembly facility via polyphasic taxonomy, Illumina genome sequencing, and draft genome assembly, followed by phylogenomics analysis. The authors report the presence of genes that can help synthesize lasso peptides by the microbes that share a high similarity with those reported here. Gene clusters that likely code for siderophores that could play a role in sequestering heavy metals from contaminated environments were also identified in these microbes. This study is interesting from the perspective of how such microbes may survive in these clean room environments. The analysis is very well defined on the wiki: <https://github.com/RamanLab/Neobacillus/wiki>. While the manuscript covers all essential aspects, I feel some points could be addressed better, as listed below:

Ans: Line by line clarifications and/or modifications are made as suggested by the reviewer.

Data availability: The metagenomics data reported in the manuscript don't seem to be available. Likewise, genomes referred to in line 139 of page 4 are also unavailable.

Ans: The 236 shotgun metagenomic sequences are released and made available through the NCBI under accession number [PRJNA1150505](https://www.ncbi.nlm.nih.gov/PRJNA1150505). Regarding the genomes mentioned in line 139, we have clarified the source in the manuscript by citing Wood et al., 2021 (Reference 3) as the origin of these genomes (See Line #141).

Page 3 lines 92-93: Here it is claimed that the two strains did not match with any known bacterial species with ANI >87%. Later in the manuscript, similarity to a couple of other strains described in Table 1 and SF1 with ANI values > 95% is mentioned. Is this because these genomes were not available in NCBI at the time of publication of ref 3?

Ans: Yes, the statement in Lines 92-93 is correct since these two strains did not match with any validly described bacterial "species" since ANI is <87% with other recognized *Neobacillus* species. As

noted by the reviewer, we recognized two other strains (not species described in Table 1 and SF1) exhibited >95% ANI values hence all four strains belong to the novel *Neobacillus drikisii* species. These other two strains mentioned in Table 1 were not from our group but their genomes were deposited in NCBI by other researchers and retrieved for this study. Also, procuring these two strains was not possible despite our attempts to contact them. So, we genomically characterized those strains based on their genomes retrieved from NCBI but strains were not available for our study. Hence, these two strains formed a clade with our two Mars 2020 strains (totalling 4 strains), which is *N. drikisii* but different from other established *Neobacillus* species. We added the following sentences in the modified manuscript to clarify this issue (Lines 157 to 163).

- The WGS of the above said two Mars 2020 strains were generated during this study. However, comparative ANI analyses of M2020 genomes with all available genomes in NCBI showed high ANI similarities with the genomes of two other strains (95.1% for the V4I25 strain and 96% for the AT2.8 strain). Hence, genomes of these two additional strains were retrieved from NCBI and included in this study for comparative genome characterization and the assembly statistics for all four novel strains are summarized in Table 1. Except for genomes, the strains V4I25 and AT2.8 were not available for this study.

Page 3 line 107: The logic of sequencing samples with and without treatment with propidium monoazide could be explained

Ans: The PMA assay and protocols were published in multiple manuscripts (see references 18, 64, 65). To clarify the use of PMA the following sentences are added in the modified manuscript in methods section (Line 680 to 694).

- The shotgun metagenome reads were not generated during this study but downloaded from the NCBI (PRJNA1150505). In this study, we focused on identifying new microbial species in regulated environments and also mapped its presence and persistence by analyzing 236 paired-end shotgun metagenomic samples from the JPL-SAF that contained reads associated with PMA treated and untreated samples. When samples were treated with PMA, the dye first intercalates with DNA subsequently on photo-activation, covalently binds DNA and hence DNA not amplified during PCR process and related molecular analyses. Since this PMA treatment step eliminates free DNA and also penetrates the cell wall of the compromised microbial cells, the resulting metagenome reads were associated with intact and viable microorganisms (See references: 19, 67, 68).

Page 4, line 139: Results section: Here, it is mentioned that WGS was carried out for 17 + 13 isolates. If this work is part of this manuscript, please provide a link to the data and mention how the isolates were characterized. Was it based on 16S or ANI? Likewise, what method was employed for the characterization of the four uncharacterized strains? Did the authors also look at AAI and PoCP in addition to ANI?

Ans: Other than the two strains of the novel species described in this paper, the genomes were published in Wood et al., 2023 (See Ref #3). In order to clear the confusion, we mentioned the reference of the previous study in the modified manuscript. The phylogeny is based on WGS and the strains that had >95% ANI are considered as the same species. We did check AAI and PoCP in addition to ANI to decide the speciation, and the results are presented in the modified Table 2.

Page 4 line 153-161: The section on molecular phylogeny and genomic relatedness is very confusing and poorly written. Table 1 lists four novel strains; however, only two of these were sequenced by this group. Also, the statistics 14 contigs and 54 scaffolds. Is this correct? What does constructed with high-quality sequences mean?

Ans: As per NCBI, contigs less than 100 were considered as high-quality genomes without “N” in the sequence. Since we don’t have circular genomes, these were considered as draft genomes but high-quality. The following sentences were added to clarify the content.

- The draft assembly of strain 179-C4-2-HS^T exhibited a high N50 value of 4.4 Mb across 14 contigs, indicative of a high-quality assembly. However, strain 179-J 1A1 HS had a N50 of 378 kb across 54 contigs, a more fragmented assembly that may require further refinement.

The following texts were added in the modified manuscript to clarify why we have had only two strains from the Mars 2020 project but included other two genomes that are not isolated during this study (See Lines 157 to 175).

- The WGS of the above said two Mars 2020 strains were generated during this study. However, comparative ANI analyses of the genomes of Mars 2020 strains with all available genomes in NCBI showed high ANI similarities with the genomes of two other strains (95.1% for V4I25 strain and 96% for the AT2.8 strain). Hence, genomes of these two additional strains were retrieved from NCBI and included in this study for comparative genome characterization. The assembly statistics for all four novel strains are summarized in Table 1. Except for genomes, the strains V4I25 and AT2.8 were not available for this study. The draft assembly of strain 179-C4-2-HS^T exhibited a high N50 value of 4.4 Mb across 14 contigs, indicative of a high-quality assembly. However, strain 179-J 1A1 HS had a N50 of 378 Kb across 54 contigs, a more fragmented assembly that may require further refinement. The genome sizes of all four strains were approximately 6.2 Mb with a GC content of 38.3%. The total number of predicted genes was 5,832 for 179-C4-2-HS^T and 5,870 for 179-J1A1-HS strains (Table 1). Table 2 presented the similarities among closely related members of the novel species based on ANI, digital DNA-DNA Hybridization (dDDH), Average Amino-acid Identity (AAI), Percentage of Conserved Proteins (PoCP) and two marker genes (16S rRNA and *gyrB*). The marker genes also showed >99.7% (16S rRNA gene) and 98.4% (*gyrB* gene) sequence similarities between the Mars 2020 isolates described during this study with the V4I25 and AT2.8 strains. Hence, all four strains that were confirmed as novel species based on the molecular phylogeny (ANI, dDDH, AAI, PoCP, 16S rRNA, *gyrB* genes) were included in the genomic comparison, even though V4I25 and AT2.8 strains were not isolated during this study.

Page 4 line 163: claims that the four strains share considerable genomic similarity with *N. niacini* with ANI of 86.4%. Yet, in the introduction similarities <87% (see line 93) don't mean considerable genomic similarity.

Ans: Clarified and modified the text as below.

- The comparative genomic analysis indicated that the four novel strains shared <86.4% ANI similarities with *N. niacini* NBRC 15566^T (=DSM 2923^T) (Supplemental File S1) and a dDDH value of 32.5% (Table 2), hence these four strains deserve a novel species status.

Page 5 lines 184-187: The rationale for selecting *hisH* and *hisZ* is not provided.

Ans: Genes encoding *bisH* and *bisZ* were chosen based on their ubiquitous occurrence in all known *Neobacillus* species. We mentioned this rationale in the Introduction (See Lines 104 to 105).

Also, the claim made in line 190 doesn't appear to be correct based on what is shown in the supplementary figures 2A and 2B.

To clarify this comment, we have modified the statement as given below (See Lines 201 to 204).

- Although *bisH* and *bisZ* trees did not exactly match with WGS tree, all four novel *N. drikisii* strains clustered together to form a distinct clade, tightly associated with the *N. niacini* (83% for *bisH* and 100% for *bisZ* bootstrap support values).

Another general comment: Since the two genomes reported here are draft in multiple contigs, a tool like contigtax could be run to ascertain no contamination.

Ans: CheckM has been utilized to assess for contamination in these genomes (Line 642). Additional 16S rRNA gene based analysis conducted with the ContEst16S tool determined that no contamination was detected for all the genomes studied.

Page 6 line 253: Data from the metagenomics sequencing are not available. Please provide links to the same.

Ans: We have made the 236 shotgun metagenomic sequences available through the NCBI under accession number PRJNA1150505.

Page 7 line 299: Are you suggesting horizontal gene transfer here?

Ans: The protein sequence of paeninodin identified in the novel *N. drikisii* described in this study exhibited 80% similarity to the paeninodin found in *P. dendritiformis*. Furthermore, as indicated in Table 1, the biosynthetic pathways of paeninodin in eight *Neobacillus* species, including *N. drikisii*, showed similarity percentages ranging from 60% to 100% with those of *P. dendritiformis*. However, additional studies are necessary to determine whether these similarities result from horizontal gene transfer or other evolutionary mechanisms.

Page 8 line 350: Have you looked for plasmid genes in the two smaller contigs of V4I25? Also, given that the genomes reported here are draft genomes any contigs that are likely to be from plasmids could be highlighted.

Ans: Following reviewer suggestions, all *N. drikisii* genomes were bioinformatically examined for the presence of plasmid borne contigs using mob-recon of mob-suite v3.1.9. Our analysis suggested that the *N. drikisii* genomes were void of plasmid borne contigs. Moreover, an analysis was carried out using additional tools, such as Plasmid-Finder-2.1 [Carattoli, *et al.* (2014.) led to the inconclusive results in other tested genomes. As neither of the tools confirmed alignment to the relaxase gene in contigs showing match to plasmid NZ_CP059409, we hence decided to modify that section in the manuscript and remove Supplementary Figure S9.

As suggested by the reviewer, we have looked into the two smaller contigs of V4I25 and confirmed that they are not plasmids.

Page 9 line 388: There is an unnecessary niaci before Apart

Ans: Removed.

Page 10 lines 441-443: Genes encoding paeninodin biosynthesis are suggested to serve as taxonomic biomarkers. However, from the analysis this claim is not justified. Please elaborate as to why these are good taxonomic biomarkers.

Ans: Authors wrote the following text about the use of paeninodin as potential taxonomic biomarker in the results section (See Lines 302 to 310).

- *N. drikisii* was differentiated from its closest phylogenetic neighbor, *N. niacini* NBRC 15566^T, by the absence of active paeninodin biosynthetic pathway in *N. niacini*. Comparative sequence analysis of paeninodin precursor peptides among all *Neobacillus* species (n=26) revealed a heterogeneous distribution of the paeninodin biosynthetic pathway in seven established *Neobacillus* species (Figure 3A and 3B; Table 2). The precursor peptides of paeninodin produced by Mars 2020 strains were also found in other environmental *N. drikisii* strains, establishing a specific lineage (Figure 3A). A clade for strains of *N. drikisii* was observed, strongly associated (95% confidence of the node) with *P. dendritiformis* C454 and *N. soli* DSM 15604^T, suggesting possible evolutionary relatedness.

Page 10 lines 460-461: Appears to be a very loose statement. Also, no discussion about the plant hormone binding genes are mentioned elsewhere in the manuscript.

Ans: In the results section (Line #342 to 343), we mentioned *interactions with plant hormones* (GO:0010011, *auxin binding*). This was further elaborated in the discussion. (Lines 475-477).

Page 11 lines 467-469: It may be worthwhile to map the reads from the various genomes to this plasmid and show how much of it is present since plasmids can vary widely in gene content even among closely related microbes.

Ans: Georgios We did scan several *Neobacillus* genomes, not all of which indicated the presence of genes similar to those in the reference plasmid database (<https://github.com/HubertTang/PLASMe>). Moreover an analysis was carried out using additional tools, such as Plasmid-Finder-2.1 [Carattoli, *et al.* (2014.) led to the inconclusive results in other tested genomes. As neither of the tools confirmed alignment to the relaxase gene in contigs showing match to plasmid NZ_CP059409, we hence decided to modify that section in the manuscript and remove Supplementary Figure S9.

Page 11 line 512: some analysis of sporulation genes in these microbes may be presented.

Ans: The following texts were added in the modified manuscript focusing on the presence of sporulation genes in these microbes

- Furthermore, the genomes of these microbes reveal the presence of several sporulation-related genes (GO:0043934), with varying numbers across strains: 9 in SAF strains (179-C4-2-HS^T, 179-J1A1-HS), 10 in V4I25, 7 in *N. niacini* NBRC 15566^T, and 6 in *N. drikisii* AT2.8.

Page 14 lines 615-627: How much data was obtained? Data coverage and genome assembly statistics could be provided.

Ans: The data coverage (33x to 728x) and genome assembly statistics and the number of filtered reads (920K to 18M reads) are provided in Table 1.

Page 14: In line 635, the outgroup is *Paenibacillus polymyxa*, and in line 646, the outgroup is *Priestia aryabhatai*. Why were different outgroups chosen?

Ans: *Paenibacillus polymyxa* is the type species of *Paenibacillus*, whereas *Priestia (Bacillus) aryabhatai* is a reclassified *Bacillus* species (formerly known as *Bacillus aryabhatai*). *Bacillus aryabhatai* was placed outside the *Neobacillus* clade based on the INDELS/associated phylogenetic analysis (See reference Patel and Gupta, 2020) and later it was reclassified as *Priestia (Bacillus) aryabhatai* (Gupta et al., 2020). *Priestia (Bacillus) aryabhatai* was a more appropriate taxa for our *hisH* and *hisZ* based analysis and hence it was taken into consideration as an outgroup.

Page 16 line 722: The wiki seems to suggest that the query was *Agrococcus*. Is this correct?
esearch to get the list of representative species accession IDs search -query '[ORGN] AND "representative genome"[filter] AND all[filter] NOT anomalous[filter]' -db assembly | esummary | xtract -pattern DocumentSummary -def "NA" -element AssemblyAccession > _repr.txt # bit to fetch the genomes bit-dl-ncbi-assemblies -w _repr.txt -f fasta -j 100

Ans: The wiki incorrectly mentioned *Agrococcus*; the intended query was for *Neobacillus*. We have updated the wiki to reflect this.

Reviewer #2 (Comments for the Author):

This manuscript reports detailed characterization of two novel strains of Gram-positive bacteria (179-C4-2-HS, 179-J1A1-HS), isolated from the Mars 2020 spacecraft assembly facility clean room, which survived heat treatment (80°C; 15 min). The results from ANI and other genomic similarity analyses show that these two strains are closely related (i.e. within the accepted species range) to the sequences of two other strains (AT2.8 from the Agave plant, and V4I25 from wheat-associated soil), whose genomes were available in the public database. This manuscript describes characterization of these four strains, based on their genome sequences, using multiple approaches. In phylogenetic trees, these strains clustered together, and they were most closely related to the species *Neobacillus niacini* NBRC 15566T. The 16S rRNA gene sequence similarity (>99.2%), average nucleotide identity (ANI) value (86.4%), *gyrB* sequence similarity (90.2%), and digital DNA-DNA hybridization values (32.5%) also supported the inference that these strains differ significantly from their closest relative *N. niacini* NBRC 15566T, and thus constitutes a novel species named *N. driksii*, within the genus *Neobacillus*.

The authors have characterized this new species regarding its phenotypic, biochemical and chemotaxonomic characteristics and have identified some properties which distinguish this species from *N. niacini* NBRC 15566T and other *Neobacillus* species. One novel aspect of *N. driksii* and 7 other *Neobacillus* species reported here is that their genomes encode the biosynthesis pathway for lasso peptides, which possess antimicrobial properties. The authors also report metagenomic analysis showing that *N. driksii* is not commonly found in NASA cleanrooms.

Comments:

The main inference from this study that the strain 179-C4-2-HS isolated from SAF clean room represents a novel species within the genus *Neobacillus*, showing close relationships to *N. niacini* NBRC 15566T, is strongly supported by the presented data. Similarly, the inference that the genomes of *N. driksii* and 7 other *Neobacillus* species contain the genes encoding the biosynthesis pathway for lasso peptides is also supported by the presented analyses. Overall, this is a detailed and presented work and I have no significant comments on it.

Minor comments:

1. In the etymology of *N. driksii*, the initial part on line 234 should be corrected as follows: *Neobacillus driksii* (*driks*'i.i. N.L. gen. n. *driksii*

Ans: Modified

2. As the authors have noted in the Introduction, the genus *Neobacillus* was circumscribed based on the shared presence of several conserved indels in protein sequences. As *N. driksii* is a new species in this genus, these distinguishing indels in this species should be examined. This can be easily done using the AppIndels.com server. <https://appindels.com/>

Ans: As stated in the Introduction, we have analyzed conserved marker genes encoding 16S rRNA, *gyrB*, imidazole glycerol phosphate synthase subunit (*hisH*), and ATP phosphoribosyltransferase regulatory subunit (*hisZ*). However, when we have used AppIndels.com as suggested by the reviewer, the server correctly predicted *Neobacillus* as genus for strains 179-C4-2-HS^T, 179-J1A1-HS, and V4I25. However, the app could not predict the genus for strain AT2.8. We ran a couple of

other established and recognized *Neobacillus* genomes in this app but it failed to recognize all of them as *Neobacillus*.

3 I have noted a few typos on Lines 740 (half instead of halt) and 742 (Medicine instead of mEdicine). The manuscript should be thoroughly checked in this regard.

Ans: The affiliation 'Centre for Integrative Biology and Systems mEdicine' (IBSE) is the expanded form, so we prefer to use 'mEdicine' instead of 'Medicine'.

We checked thoroughly for any spelling mistakes and errors.

Re: Spectrum01376-24R1 (*Neobacillus driksii* sp. nov. isolated from a Mars 2020 spacecraft assembly facility and genomic potential for lasso peptide production in *Neobacillus*)

Dear Dr. Kasthuri Venkateswaran,

Thank you for submitting your revised manuscript to Microbiology Spectrum. We have received comments from one of the reviewers regarding your submitted manuscript; the second reviewer is already positive about it. The First reviewer has recommended additional experiments, explanations, and significant concerns. Therefore, I invite you to revise your manuscript in light of the referee's comments. Your manuscript needs major revisions, and the referee's remarks are appended below for your attention. Please ensure that the added explanations and comments in the rebuttal letter are well explained throughout the text and supported by clearly described methods.

Revision Guidelines

Sincerely,
Gaurav Sharma, Ph.D.
Editor, Microbiology Spectrum

Assistant Professor, IIT Hyderabad, India
Lab Page: <https://sites.google.com/view/sharmaglab/>

Reviewer #1 (Comments for the Author):

The authors have satisfactorily addressed most of the comments I raised earlier. Nevertheless, a few more points need clarification and addressing:

a) Since the authors have run AAI and PoCP for the genomes, the same can be mentioned in the abstract (lines 50-53 in marked up manuscript) and in all other places where ANI and other molecular marker data is used for justifying the novelty of these microbes (for example lines 174-180; lines 190-200 in marked up manuscript).

b) In Fig S6 legend there is a typographic error in the spelling of phosphor.

c) In Fig S6, the authors should mention if the same TLC plate was used for panels B and C, the order in which the plate was treated with the reagents and if this is accepted practice. Also, the Methods section should contain these details.

d) Genome assembly details from Table 1 do not match that reported in NCBI

<https://www.ncbi.nlm.nih.gov/nucore/JAROBZ000000000> is an assembly that contains 31 contigs. The data coverage reported is 438x and the sequencing technology is Illumina NovaSeq. The assembler is SPAdes v. 3.11.1.

Further,

<https://www.ncbi.nlm.nih.gov/Taxonomy/Browser/wwwtax.cgi?mode=Undef&id=2675272&lvl=3&keep=1&srchmode=1&unlock> does not contain link to 179-J 1A1 HS genome. Also, <https://www.ncbi.nlm.nih.gov/nucore?term=JBDZYE000000000> is a dead link.

Given that the data corresponds to a type strain of a newly defined species from a restricted environment, namely a NASA mission assembly cleanroom, it is imperative that the manuscript presents the most accurate and comprehensive assembly and analysis possible. Of the two genomes presented in the manuscript, only one assembly is publicly accessible on NCBI. This assembly seems to be based only on Illumina data and does not correspond to that reported in Table 1, which indicates a hybrid assembly using both Illumina and Oxford Nanopore data.

It is surprising that even with ample Illumina and Nanopore data (I am estimating that there is good ONT data coverage based on the difference in average coverage provided in Table 1 vs that on NCBI), presumably exceeding 100x coverage, the authors were unable to generate a complete genome assembly. Since no details of the hybrid assembly are provided in the methods section, I am not certain how this was done. My guess is that it was also using SPAdes?

I recommend that the authors make the Illumina and ONT read data available on SRA and/or on the GitHub wiki. They could explore hybrid assembly using Unicycler, and long-read-first approaches followed by polishing using the Illumina data using assemblers such as Tricycler, Dragonflye or Hybracter (this may also be helpful for plasmids if any). Furthermore, considering that most of the analysis after the genome assembly is available on the provided GitHub wiki page, I suggest they also include the data and assembly details of the genomes there for transparency and reproducibility.

Re: Spectrum01376-24R1 (*Neobacillus driksii* sp. nov. isolated from a Mars 2020 spacecraft assembly facility and genomic potential for lasso peptide production in *Neobacillus*)

Dear Dr. Kasthuri Venkateswaran,

Thank you for submitting your revised manuscript to Microbiology Spectrum. We have received comments from one of the reviewers regarding your submitted manuscript; the second reviewer is already positive about it. The First reviewer has recommended additional experiments, explanations, and significant concerns. Therefore, I invite you to revise your manuscript in light of the referee's comments. Your manuscript needs major revisions, and the referee's remarks are appended below for your attention. Please ensure that the added explanations and comments in the rebuttal letter are well explained throughout the text and supported by clearly described methods.

Ans: We appreciate the valuable feedback provided by both reviewers and are pleased to hear that one of the reviewers is positive about the manuscript. We have taken into consideration the further suggestions of the reviewer and have addressed them by duly incorporating them into our revised manuscript.

Reviewer #1 (Comments for the Author):

The authors have satisfactorily addressed most of the comments I raised earlier. Nevertheless, a few more points need clarification and addressing:

a) Since the authors have run AAI and PoCP for the genomes, the same can be mentioned in the abstract (lines 50-53) and in all other places where ANI and other molecular marker data is used for justifying the novelty of these microbes (lines 174-180; 190-200).

Ans: Necessary modifications have been made in the manuscript.

b) In Fig S6 legend there is a typographic error in the spelling of phosphor.

Ans: Modified.

c) In Fig S6, the authors should mention if the same TLC plate was used for panels B and C, the order in which the plate was treated with the reagents and if this is accepted practice. Also, the Methods section should contain these details.

Ans: Yes. The same TLC plate (panels B and C) was used for staining amino and phospholipids. This is an accepted practice and standard procedure for polar lipid analysis (Minnikin *et al.* 1984).

- The TLC plate was stained first with ninhydrin reagent to locate aminolipids (pink spots). Subsequently, the same TLC plate was used to stain with Dittmer and Lester's Zinzadze reagent, in which pink spots of amino lipid vanish and blue spots of phospholipids emerge. Total lipids and glycolipids required additional two TLC plates.

A valid reference for polar lipids has already been included (Minnikin *et al.*, 1984). In addition, staining reagents have been specified in the methods part (lines 638-642).

d) Genome assembly details from Table 1 do not match that reported in NCBI

<https://www.ncbi.nlm.nih.gov/nuccore/JAROBZ000000000> is an assembly that contains 31 contigs. The data coverage reported is 438x and the sequencing technology is Illumina NovaSeq. The assembler is SPAdes v. 3.11.1.

Further,

<https://www.ncbi.nlm.nih.gov/Taxonomy/Browser/wwwtax.cgi?mode=Undef&id=2675272&vl=3&keep=1&srchmode=1&unlock>

does not contain link to 179-J 1A1 HS genome. Also, <https://www.ncbi.nlm.nih.gov/nuccore?term=JBDZYE000000000> is a dead link.

Given that the data corresponds to a type strain of a newly defined species from a restricted environment, namely a NASA mission assembly cleanroom, it is imperative that the manuscript presents the most accurate and comprehensive assembly and analysis possible. Of the two genomes presented in the manuscript, only one assembly is publicly accessible on NCBI. This assembly seems to be based only on Illumina data and does not correspond to that reported in Table 1, which indicates a hybrid assembly using both Illumina and Oxford Nanopore data.

Ans: The recent genome assemblies for 179-C4-2-HS^T and 179-J1A1-HS have been released and are now available on NCBI. The updated assembly details for JAROBZ000000000 now match the reported information in Table - 1.

179-C4-2-HS^T: [JAROBZ000000000](https://www.ncbi.nlm.nih.gov/nuccore/JAROBZ000000000)

179-J1A1-HS: [JBDZYE000000000](https://www.ncbi.nlm.nih.gov/nuccore/JBDZYE000000000)

It is surprising that even with ample Illumina and Nanopore data (I am estimating that there is good ONT data coverage based on the difference in average coverage provided in Table 1 vs that on NCBI), presumably exceeding 100x coverage, the authors were unable to generate a complete genome assembly. Since no details of the hybrid assembly are provided in the methods section, I am not certain how this was done. My guess is that it was also using SPAdes?

Ans: We performed the hybrid assembly using Unicycler with both Illumina and Nanopore data. The methods section has been updated to reflect this (lines 664-673). Despite best practices and high coverage, extremophilic Gram-positive genomes can be challenging to complete due to high GC content, extensive repetitive regions as well as unusual DNA modifications (e.g., methylation patterns) leading to unresolved gaps in the assembly.

I recommend that the authors make the Illumina and ONT read data available on SRA and/or on the GitHub wiki. They could explore hybrid assembly using Unicycler, and long-read-first approaches followed by polishing using the Illumina data using assemblers such as Tricycler, Dragonflye or Hybracter (this may also be helpful for plasmids if any). Furthermore, considering that most of the analysis after the genome assembly is available on the provided GitHub wiki page, I suggest they also include the data and assembly details of the genomes there for transparency and reproducibility.

Ans: Necessary genome assembly steps have been added to the GitHub wiki page: <https://github.com/RamanLab/Neobacillus/wiki>

Both the Illumina and ONT reads are submitted to NCBI and released on Oct 1, 2024 and will be made available under the given accessions after NCBI curation.

Illumina reads (SUB14482789): 179-C4-2-HS^T: [SRR29255618](https://www.ncbi.nlm.nih.gov/sra/SRR29255618) ; 179-J1A1-HS: [SRR29255617](https://www.ncbi.nlm.nih.gov/sra/SRR29255617)

ONT reads: 179-C4-2-HS^T: [SRR30855205](https://www.ncbi.nlm.nih.gov/submit/submit.cgi?acc=SRR30855205): The NCBI portal shows it is released on Oct 1, 2024 and usually it will take one week before it is made public. See the screen shot below.

Submission Portal Home My submissions **Manage data** Groups Templates My profile

Manage Data > SRA: SRR30855205

BioProject: PRJNA935338 Novel bacterial species from MT1 and MT2 ISS missions
 BioSample: SAMN43944611 Neobacillus driksii
 SRR30855205 WGS of Neobacillus driksii 179-J C42 HS

SRA accession: SRR30855205

Status: ✔ Released

Release date: 2024-10-01

Created: 2024-10-01 13:17

Updated: 2024-10-01 13:30

Experiment information	Experiment	Library ID	Library strategy	Library source	Library selection	Library layout	Platform	Instrument
	SRX26253578	179-J C42 HS ONT	WGS	GENOMIC	RANDOM	SINGLE	OXFORD NANOPORE	PromethION

Experiment description: Samples were collected from the floor of the Spacecraft Assembly Facility at the Jet Propulsion Laboratory. Samples were collected from 1 m2 of the floor using a 23x23 cm premoistened polyester wipe, placed in a glass bottle containing phosphate buffer saline, thoroughly mixed for 30 s, and concentrated using 0.45 m contracting pipette tip (InnovaPrep, Drexel, MO, USA). An aliquot of 425 L was heat shocked (80C, 15 min), and grown on tryptic soy
Show more...

File information	File name	File type	MD5SUM
	179-J_C42_HS_ONT.fastq	fastq	bd00c7ee85047f6fbdaed04a83ae4a51

Statistics	Total number of spots	Total number of bases	GC percentage
	64000	228456802	38.63

Taxonomy analysis	Organism	Percent aligned
	Neobacillus sp. 179-C4.2 HS	80.79
	Escherichia coli	2.60
	Neobacillus niacini	0.10
	Neobacillus sp. DY30	0.00

Re: Spectrum01376-24R2 (*Neobacillus driksii* sp. nov. isolated from a Mars 2020 spacecraft assembly facility and genomic potential for lasso peptide production in *Neobacillus*)

Dear Dr. Kasthuri Venkateswaran,

Based on two revisions and associated quality improvement, I am pleased to inform you of the acceptance of your manuscript entitled "Neobacillus driksii sp. nov. isolated from a Mars 2020 spacecraft assembly facility and genomic potential for lasso peptide production in *Neobacillus*" in its current form for publication in Microbiology Spectrum.

I am forwarding it to the ASM production staff for publication. Your paper will first be checked to make sure all elements meet the technical requirements. ASM staff will contact you if anything needs to be revised before copyediting and production can begin. Otherwise, you will be notified when your proofs are ready to be viewed.

Sincerely,
Gaurav Sharma, Ph.D.
Editor, Microbiology Spectrum

Assistant Professor, IIT Hyderabad, India
Webpage: <https://sites.google.com/view/sharmaglab/>